# Frequency switching between oscillatory homeostats and the regulation of p53

**Peter Ruoff**[1]*, **Nobuaki Nishiyama**[2]

**1** Department of Chemistry, Bioscience, and Environmental Engineering, University of Stavanger, Stavanger, Norway, **2** Division of Mathematical and Physical Sciences, Graduate School of Natural Science and Technology, Kanazawa University, Kanazawa, Japan

* peter.ruoff@uis.no

**Data Availability Statement:** All relevant data are within the manuscript and its Supporting Information files.

**Funding:** The authors received no specific funding for this work.

## Abstract

Homeostasis is an essential concept to understand the stability of organisms and their adaptive behaviors when coping with external and internal assaults. Many hormones that take part in homeostatic control come in antagonistic pairs, such as glucagon and insulin reflecting the inflow and outflow compensatory mechanisms to control a certain internal variable, such as blood sugar levels. By including negative feedback loops homeostatic controllers can exhibit oscillations with characteristic frequencies. In this paper we demonstrate the associated frequency changes in homeostatic systems when individual controllers -in a set of interlocked feedback loops- gain control in response to environmental changes. Taking p53 as an example, we show how Per2, ATM and Mdm2 feedback loops -interlocked with p53- gain individual control in dependence to the level of DNA damage, and how each of these controllers provide certain functionalities in their regulation of p53. In unstressed cells, the circadian regulator Per2 ensures a basic p53 level to allow its rapid up-regulation in case of DNA damage. When DNA damage occurs the ATM controller increases the level of p53 and defends it towards uncontrolled degradation, which despite DNA damage, would drive p53 to lower values and p53 dysfunction. Mdm2 on its side keeps p53 at a high but sub-apoptotic level to avoid premature apoptosis. However, with on-going DNA damage the Mdm2 set-point is increased by HSP90 and other p53 stabilizers leading finally to apoptosis. An emergent aspect of p53 upregulation during cell stress is the coordinated inhibition of ubiquitin-independent and ubiquitin-dependent degradation reactions. Whether oscillations serve a function or are merely a by-product of the controllers are discussed in view of the finding that homeostatic control of p53, as indicated above, does in principle not require oscillatory homeostats.

## Introduction

The concept of homeostasis is central to our understanding how organisms and cells adapt to their environments and thereby maintain their stability [1–3]. With the development of cybernetics [4, 5] control engineering concepts were, for the first time, applied to biological systems [6, 7]. With the advancement of molecular biology, robust control theoretic methods were

**Competing interests:** The authors have declared that no competing interests exist.

applied at the molecular level, such as integral reign control [8], alongside with integral feedback [9–11], and systems biology methods [12, 13]. To achieve robustness of feedback controllers by integral control it became clear that certain reaction kinetic conditions need to be met. These conditions include zero-order kinetics [9, 10, 14–19], autocatalysis [20–22], and second-order (bimolecular/antithetic) reactions [23, 24], which were implemented into various controller motifs, synthetic gene networks, and other negative feedback structures [18, 25–27].

A particular interesting aspect is that, under certain conditions, the homeostatic controllers may become oscillatory and preserve, if integral control is present, their homeostatic property by keeping the *average value* of the controlled variable at its set-point [28]. While the occurrence of oscillations is generally avoided in control engineering, oscillatory behavior is ubiquitously found in natural systems, exemplified by the adaptive properties of circadian and ultradian rhythms [29–31].

In this paper we show how a set of inter-connected negative feedback loops maintain robust homeostasis in a controlled variable both under non-oscillatory and oscillatory conditions. We show that oscillatory controllers (negative feedback loops) may have specific frequencies and that frequency switching between different controllers occur dependent on the perturbation level of the controlled variable. We demonstrate how three combined negative feedback structures (see Materials and methods) reflect aspects of p53 regulation by involving the proteins Per2, ATM, and Mdm2. Dependent whether p53 degradation or synthesis is dominant, and dependent whether controllers are oscillatory, either low-level p53 circadian rhythms or higher-level p53 ultradian oscillations can be observed. However, whether oscillations serve a function or are merely a by-product of an oscillatory nature of the controllers is discussed in view of the finding that homeostatic control of p53 does in principle not require oscillatory homeostats.

## Materials and methods

Computations were performed by using the Fortran subroutine LSODE [32] and Matlab (mathworks.com). Plots were generated with gnuplot (gnuplot.info) and edited with Adobe Illustrator (adobe.com). To make notations simpler, concentrations of compounds are denoted by compound names without square brackets and generally given in arbitrary units (au). Time derivatives are indicated by the 'dot' notation. Rate parameters are generally presented as $k_i$'s ($i = 1, 2, 3, \ldots$) irrespective of their kinetic nature. However, some $K_M$'s (Michaelis constants), $K_I$'s (inhibition constants), and $K_a$'s (activation constants) are emphasized when they are considered to play a role in the oscillatory and regulatory behavior of controllers. In the Supporting Information a set of Matlab (S1 Matlab), gnuplot, and avi-files (S1 & S2 Gnuplots) are provided to illustrate results.

### The controller motifs used in this study

Drengstig et al. [18] suggested a basic set of eight 2-species negative feedback structures (termed controller motifs) with the incorporation of integral control. When analyzing the feedback structures between p53 and Per2, ATM, and Mdm2 we found that the three p53 feedback loops matched with the structures of motifs m3, m1, and m5, respectively (see more below).

The motifs (m1, m3, m5) are shown in Fig 1 *A* and $E_i$ are the controlled and controller species, respectively. When integral control is invoked, the rate equations of the $E_i$'s (see below) define the different controllers' set-points of *A*. Motifs m1 and m3 are termed inflow controllers. They oppose outflow perturbations of *A* by the $E_i$-induced compensatory reactions. Motif m5 is an outflow controller which opposes *A*-increasing perturbations by removing *A* due to $E_5$.

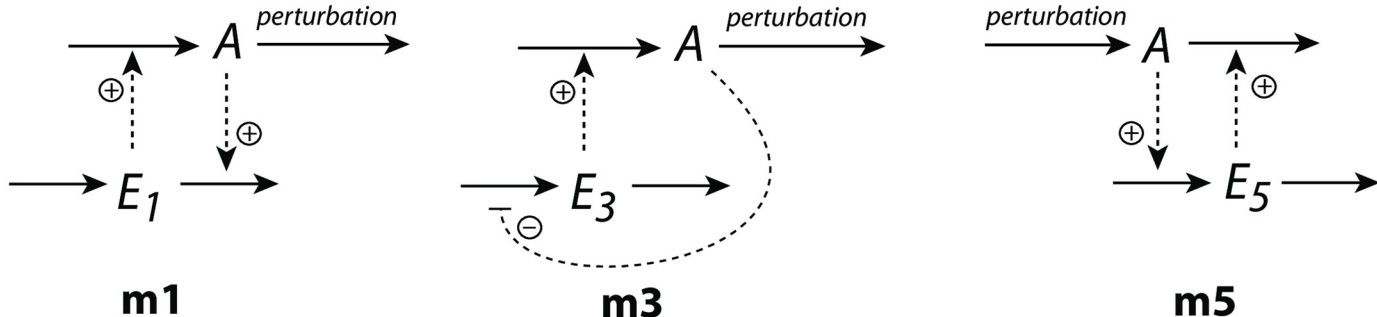

**Fig 1. The negative feedback loops m1, m3, and m5 used in this study.** Solid arrows indicate reactions/mass flow, while dashed lines indicate activating or inhibiting signaling events. *A* is the controlled variable, which is kept at a certain set-point by opposing (step-wise) perturbations in *A*. The $E_i$'s are controller species.

Rate constants have been arbitrarily chosen, but with the constraint that period lengths of the m3 and m5 oscillators are in the range of experimentally observed values, i.e. ca. 24h for the m3 (circadian Per2) oscillator and about 5h for the m5 (Mdm2) oscillator. The other used constraint concerns the chosen set-points for *A* of the three controllers. Set-points are chosen such that wind-up between controllers is avoided. See the chapter *Synergy condition for coupled feedback loops* later in the paper discussing this point.

## Outline of the paper

In the first part of *Results and discussion* we describe how robust homeostasis can be achieved by a combination of inflow and outflow controllers using motifs m3 and m5 as an example. We show that controllers can operate both in a non-oscillatory and oscillatory control mode. In oscillatory mode, the individual controllers have, dependent on certain rate constants, characteristic inherent frequencies. When controllers become interlocked frequency changes will occur when during a perturbation a controller with a different frequency becomes dominant.

In the second part (starting with section *p53 regulation by inflow and outflow control*) we suggest how p53 in response to different stress-levels is homeostatically regulated by three interlocked oscillatory feedback loops which involve Per2 (no stress, motif m3), ATM* (medium stress, motif m1), and Mdm2 (high stress, motif m5) with the observed frequency/period changes from Per2-based circadian rhythms to m3/m5-based ultradian oscillations.

## Results and discussion

### Cannon's definition of homeostasis and its realization by inflow and outflow controllers

**The non-oscillatory case.** Cannon defined homeostasis as the result of coordinated physiological processes, which maintain most of the steady states in organisms by keeping them within narrow limits [33]. One of the typical examples are human blood calcium levels, which, throughout our lifetimes are kept between approximately 9 to 10 mg Ca per dl blood. When levels are outside that range serious illness or death may occur.

The combination of inflow and outflow controllers having integral control [18] allows to keep a regulated variable within such strict limits. Fig 2 shows the arrangement between two collaborative controllers where the set-point of the inflow controller ($A_{set}^{in}$) ensures for the lowest tolerable concentration of the controlled variable *A*, while the outflow controller does not

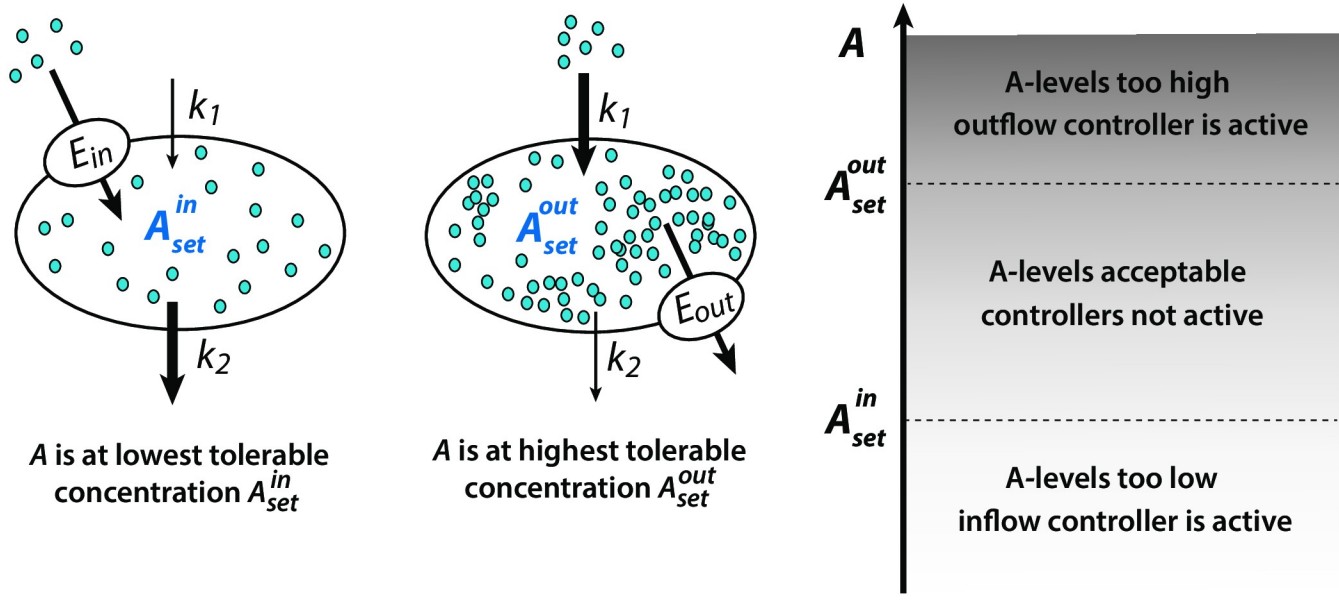

**Fig 2. Combination of inflow/outflow controllers (indicated by the transporters $E_{in}$ and $E_{out}$) which keep a controlled variable $A$ within the controllers'
set-points, independent of the perturbation parameters $k_1$ and $k_2$.**

allow that $A$-levels exceed the outflow controller's set-point $A_{set}^{out}$. It should be pointed out that
not all inflow/outflow controller combinations [18] will lead to a set of collaborative controller
pairs, because set-point values and the individual controllers' on/off characteristics need to
match; otherwise the controllers may work against each other and integral windup may be
encountered [18], as described in more detail below.

Fig 3 shows combined controller motifs m3 and m5 [18]. Feedback structure m3 is an
inflow controller, while scheme m5 is an outflow controller where $A$ is the controlled variable
and $k_1$ and $k_2$ represent perturbations. To emphasize the inflow-outflow structure of the com-
bined controllers $E_5$ and $E_3$ (Fig 1) are written as $E_{out}$ and $E_{in}$, respectively.

There are two separate conditions which have been applied on the controllers. One con-
cerns the accuracy of the implemented integral control [14, 18] by using zero-order or near
zero-order degradation/removal kinetics for the controller species $E_i$. This accuracy condition
for integral control is independent of whether the controllers are oscillatory or not.

The other condition concerns the controllers' oscillatory or non-oscillatory behaviors. When
the degradation/removal reactions of $A$ are first-order with respect to $A$ the system is non-oscil-
latory. On the other hand, when degradations of $A$ turn into zero-order kinetics with respect to
$A$, the controllers become oscillatory without loosing the integral control part [16, 28].

For the non-oscillatory case the rate equation of $A$ is:

$$\dot{A} = k_1 - k_2 \cdot A + k_6 \cdot E_{in} - k_7 \cdot A \cdot E_{out} \tag{1}$$

$E_{out}$ and $E_{in}$ have the rate equations:

$$\dot{E_{out}} = k_3 \cdot A - \frac{k_4 \cdot E_{out}}{k_5 + E_{out}} \tag{2}$$

$$\dot{E_{in}} = \frac{k_8 \cdot K_{I1}}{K_{I1} + A} - \frac{k_9 \cdot E_{in}}{k_{10} + E_{in}} \tag{3}$$

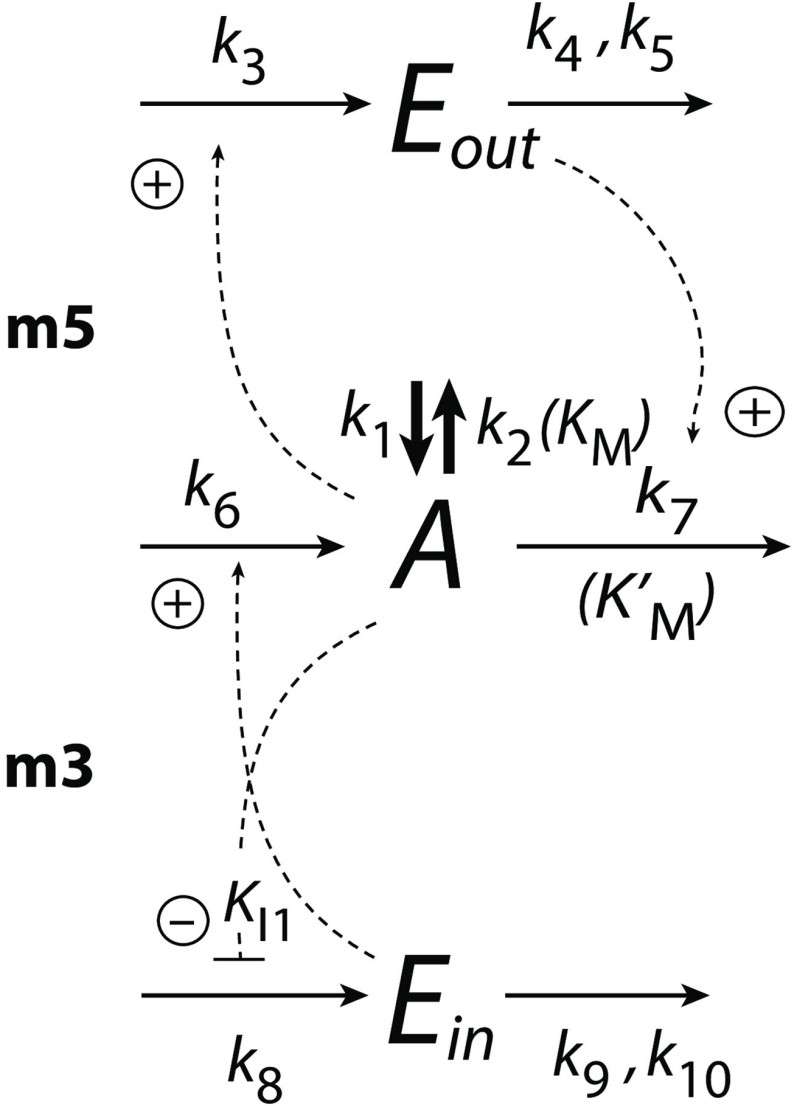

**Fig 3. Combination of controller motifs m3 and m5 with $A$ as the controlled variable.** Rate parameters $k_1$ and $k_2$ are perturbations. $K_M$ and $K_M'$ (in parentheses) are used when zero-order degradations with respect to $A$ are studied and controllers become oscillatory (see next section). $K_{I1}$ is an inhibition constant.

High accuracy of the controllers is achieved when $E_{out}$ and $E_{in}$ are removed by zero-order (or near zero-order) kinetics, i.e. the rate parameters $k_5$ and $k_{10}$ in Eq 2 (m5 controller) and Eq 3 (m3 controller) satisfy the conditions $k_5 \ll E_{out}$ and $k_{10} \ll E_{in}$.

The controllers' set-points, $A_{set}^{out}$ (for m5), and $A_{set}^{in}$ (for m3), are calculated by setting $\dot{E}_{out}$ and $\dot{E}_{in}$ to zero. Assuming $k_5 \ll E_{out}$ and $k_{10} \ll E_{in}$, we get:

$$A_{set}^{out} = A_{set}^{m5} \cong \frac{k_4}{k_3} \qquad (4)$$

$$A_{set}^{in} = A_{set}^{m3} \cong \frac{K_{I1}(k_8 - k_9)}{k_9} \qquad (5)$$

Fig 4 shows the steady state values of $A$, $E_{out}$, and $E_{in}$ as a function of the perturbation parameters $k_1$ and $k_2$. It shows that the combined controllers in Fig 3 can keep variable $A$ between the set-points of the m3 and m5 controllers. In Fig 4a the red color indicates the $A$ values that are close to or at the set-point of the outflow controller m5 ($A_{set}^{out}$), while the purple color shows the $A$ values close to or at set-point for inflow controller m3, $A_{set}^{in}$. Note the corresponding up- and downregulation of $E_{out}$ and $E_{in}$ in Fig 4b.

**Oscillatory control mode.** When the degradation reactions become zero-order with respect to $A$ the rate equation of $A$ becomes

$$\dot{A} = k_1 - \frac{k_2 \cdot A}{K_M + A} + k_6 \cdot E_{in} - \frac{k_7 \cdot A \cdot E_{out}}{K_M' + A} \tag{6}$$

with $K_M, K_M' \ll A$. In this case both the m3 and the m5 controllers become oscillatory. For the m5 feedback loop the oscillations can approximately be described by a harmonic oscillator [16] with estimates of period length $P_{m5}$ and amplitudes $A_{ampl}$, $E_{out}^{ampl}$ as

$$P_{m5} = \frac{2\pi}{\sqrt{k_3 k_7}} \tag{7}$$

$$A_{ampl} = 2 \times \sqrt{\frac{H_{5,0} + \alpha}{c}} \tag{8}$$

$$E_{out}^{ampl} = 2 \times \sqrt{\frac{H_{5,0} + \alpha}{a}} \tag{9}$$

where $a = 0.5 \times k_7$, $b = k_1 + k_6 - k_2$, $c = 0.5 \times k_3$, $d = k_4$, and $\alpha = (d^2/4c) + (b^2/4a)$. Due to a misprint in Ref [16] for the harmonic oscillator solution of controller m5, S1 Text gives the derivations of Eqs 7–9.

Also for the m3 feedback loop a "harmonic oscillator approximation" [28] can be found for the period $P_{m3}$ with semi-analytic expressions for the amplitudes. Dependent whether one starts to calculate $\ddot{A}$ or $\ddot{E}_{in}$ in deriving $P_{m3}$, two interrelated expressions for $P_{m3}$ are obtained, respectively:

$$P_{m3} = \frac{2\pi}{\sqrt{k_6 k_8 K_{I1}}} (K_{I1} + <A>) \tag{10}$$

or

$$P_{m3} = \frac{2\pi}{\sqrt{\left(\frac{k_6 k_9}{K_{I1} + <A>}\right)}} \tag{11}$$

where $<A>$ is the average value of $A$ defined as

$$<A> = \frac{1}{\tau} \int_0^\tau A(t) \, dt \tag{12}$$

See S1 Text for details.

Fig 5 shows a comparison between combined controllers m3 and m5 when using for $A$ Eq 1 (Fig 5a) and when using Eq 6 (Fig 5b). In phase 1 the outflow perturbation $k_2$ is largest ($k_1 = 1.0$, $k_2 = 10.0$) while in phase 2 the inflow perturbation is largest ($k_1 = 10.0$, $k_2 = 0.0$). Rate constant values have been chosen such that during phase 1 the dominating inflow controller (m3)

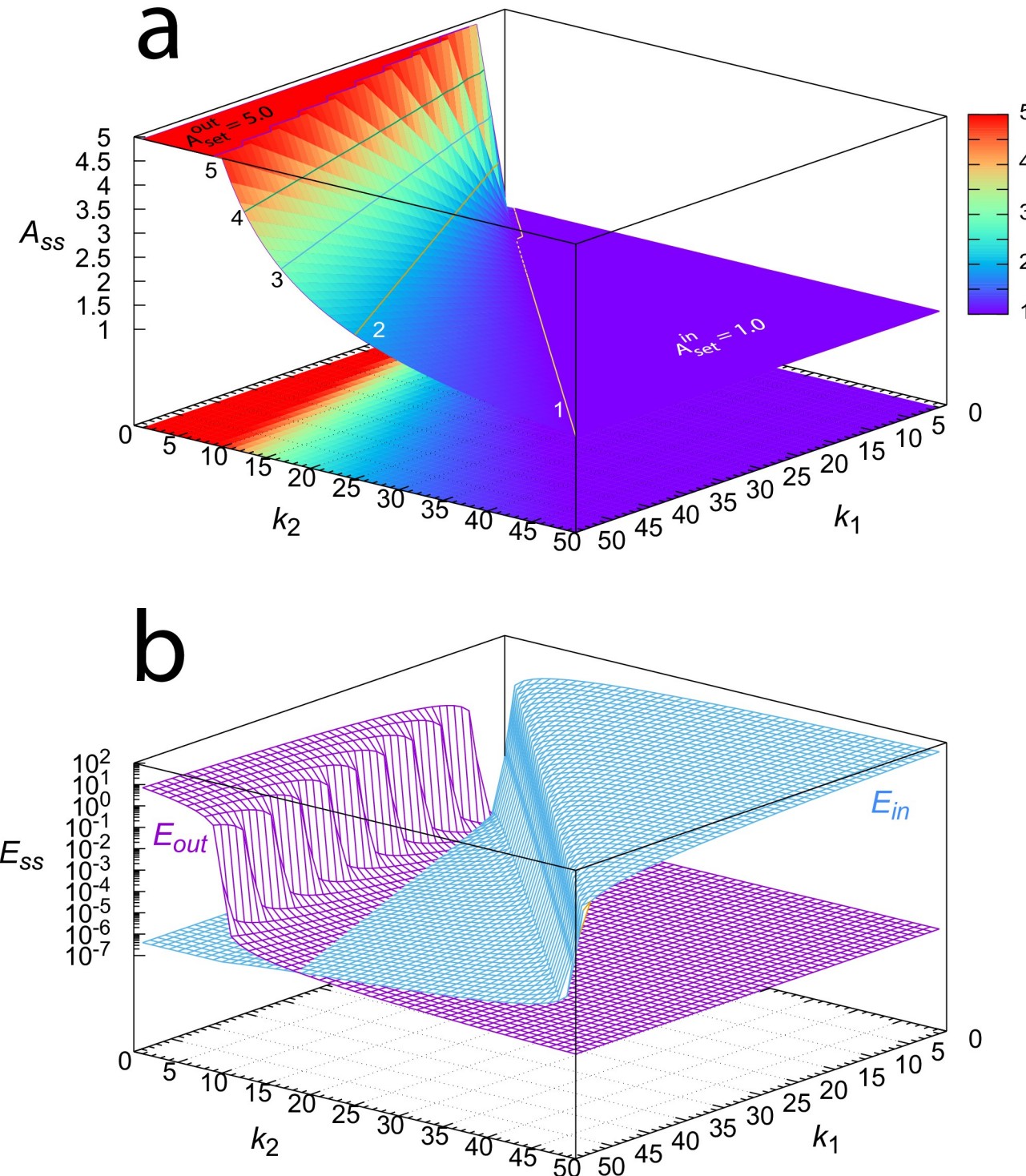

**Fig 4. Steady state values of $A$, $E_{out}$, and $E_{in}$ of the combined m3-m5 controllers (Fig 3) as a function of perturbation parameters $k_1$ and $k_2$. $k_1$ and $k_2$ vary between 1.0 and 50.0 with increments of 1.0.** (a) Steady state values of $A$. Numbers 1-5 in the plot indicate the contour lines having this value of $A_{ss}$. (b) Steady state values of $E_{out}$ and $E_{in}$. Rate constants: $k_3 = 10.0$, $k_4 = 50.0$, $k_5 = 1 \times 10^{-6}$, $k_6 = k_7 = 1.0$, $k_8 = 2.0$, $k_9 = 1.0$, $k_{10} = 1 \times 10^{-6}$, $K_{I1} = 1.0$. Initial concentrations when calculating $A_{ss}$ for each $k_1$, $k_2$ pair: $A_0 = 3.0$, $E_{in} = E_{out} = 0.0$; integration time: 1000 time units. For an interactive visualization of the surfaces, see S1 Gnuplot.

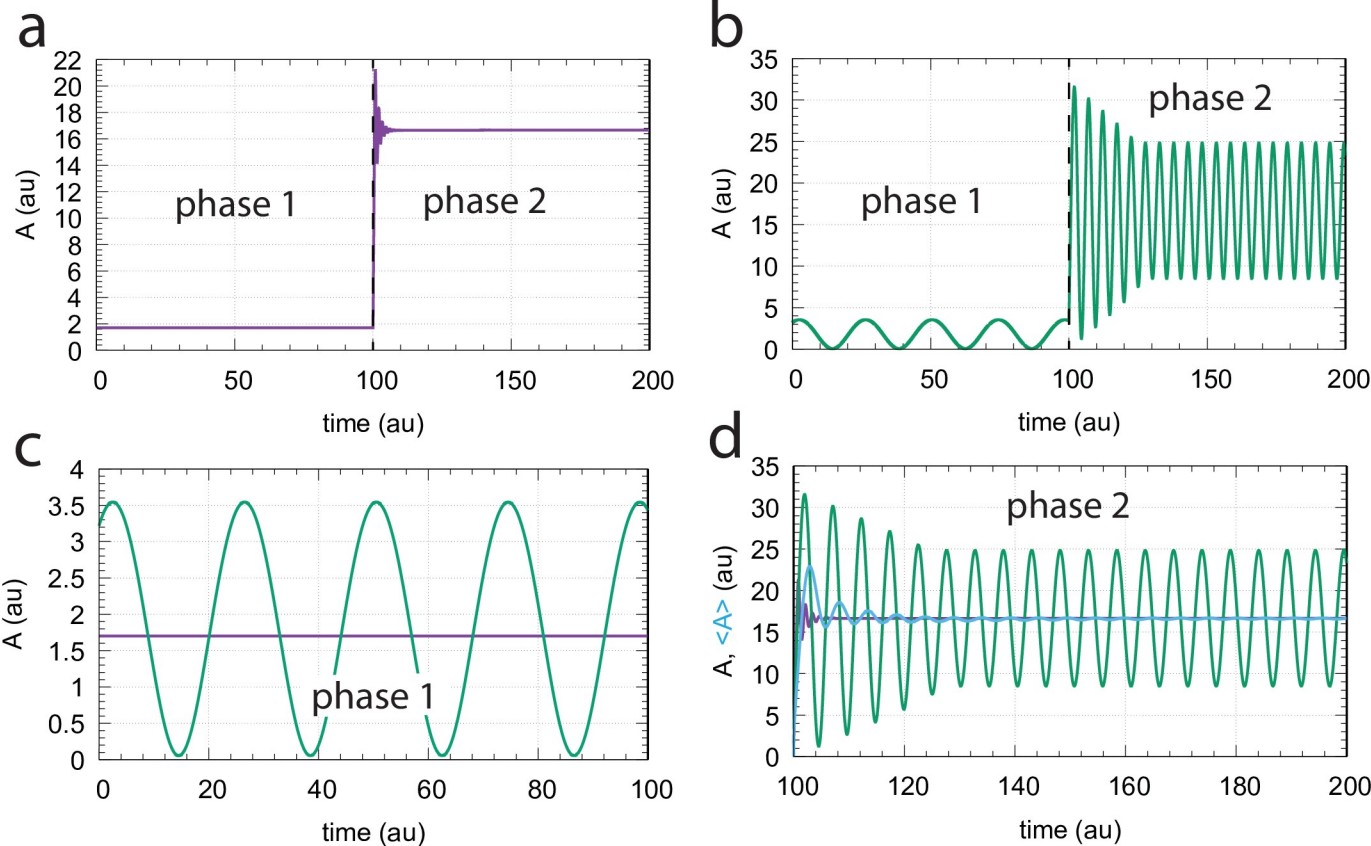

**Fig 5. Comparison between oscillatory and non-oscillatory controller modes of combined motifs m3 and m5.** (a) non-oscillatory behavior when rate of $A$ is given by Eq 1. (b) Oscillatory behavior when describing $\dot{A}$ by Eq 6. (c) Comparison between oscillatory and non-oscillatory behavior in phase 1 when $k_1 = 1.0$ and $k_2 = 10.0$. Average value of oscillatory $A$, $<A>$, is precisely at $A_{set}^{in} = 1.7$. (d) Comparison between oscillatory and non-oscillatory behavior in phase 2 when $k_1$ and $k_2$ have changed to respectively 10.0 and 0.0. $<A>$ (blue line), approaches rapidly the set-point of the outflow controller $A_{set}^{out} = 16.7$. Other rate constants: $k_3 = 3.0$, $k_4 = 50.0$, $k_5 = 1 \times 10^{-8}$, $k_6 = 0.7$, $k_7 = 0.5$, $k_8 = 1.2$, $k_9 = 1.0$, $k_{10} = 1 \times 10^{-6}$, $K_M$, $K_M'$ (when applied) both $1 \times 10^{-6}$, and $K_{I1} = 8.5$. Initial concentrations for phase 1, panel a: $A_0 = 1.700$, $E_{out,0} = 1.136 \times 10^{-9}$, $E_{in,0} = 2.286 \times 10^1$; Initial concentrations for phase 1, panel b: $A_0 = 3.219$, $E_{out,0} = 2.395 \times 10^{-9}$, $E_{in,0} = 1.322 \times 10^1$.

has a period of 24 time units (Fig 5c), while during phase 2 the dominating outflow controller (m5) has a period of approximately 5 time units (Fig 5d). These rate constant values also take part in defining the set-point for the inflow controller m3 to $A_{set}^{in} = 1.7$ (Eq 5) and the set-point of the outflow controller m5 to $A_{set}^{out} = 16.7$ (Eq 4).

In the oscillatory control mode (Eq 6) the period of the dominant (ruling) controller is established (Fig 6a), dependent whether inflow perturbation $k_1$ or outflow perturbation $k_2$ dominates. The $A$, $E_{in}$ amplitudes of the inflow controller m3 are practically constant and independent of the level of perturbation ($k_2$), while for outflow controller m5 amplitudes increase with increasing perturbation strength $k_1$ (Fig 6b). S1 Text gives approximative analytical expressions for the m3 and m5 oscillators' amplitudes. Interestingly, oscillations stop when $k_1$ and $k_2$ values are equal. Fig 6c shows that the oscillatory controllers follow (on average) closely the controllers' set-points of the non-oscillatory state (Eqs 4 and 5). Fig 6d shows the changes of the average values of respectively $E_{in}$ and $E_{out}$ ($<E_{in}>$ or $<E_{out}>$) in dependence of $k_1$ and $k_2$.

**p53 regulation by inflow and outflow control.** p53 is a protein often described as the "guardian of the genome" [34]. p53 takes part in cell fate decisions [35] with respect to internal

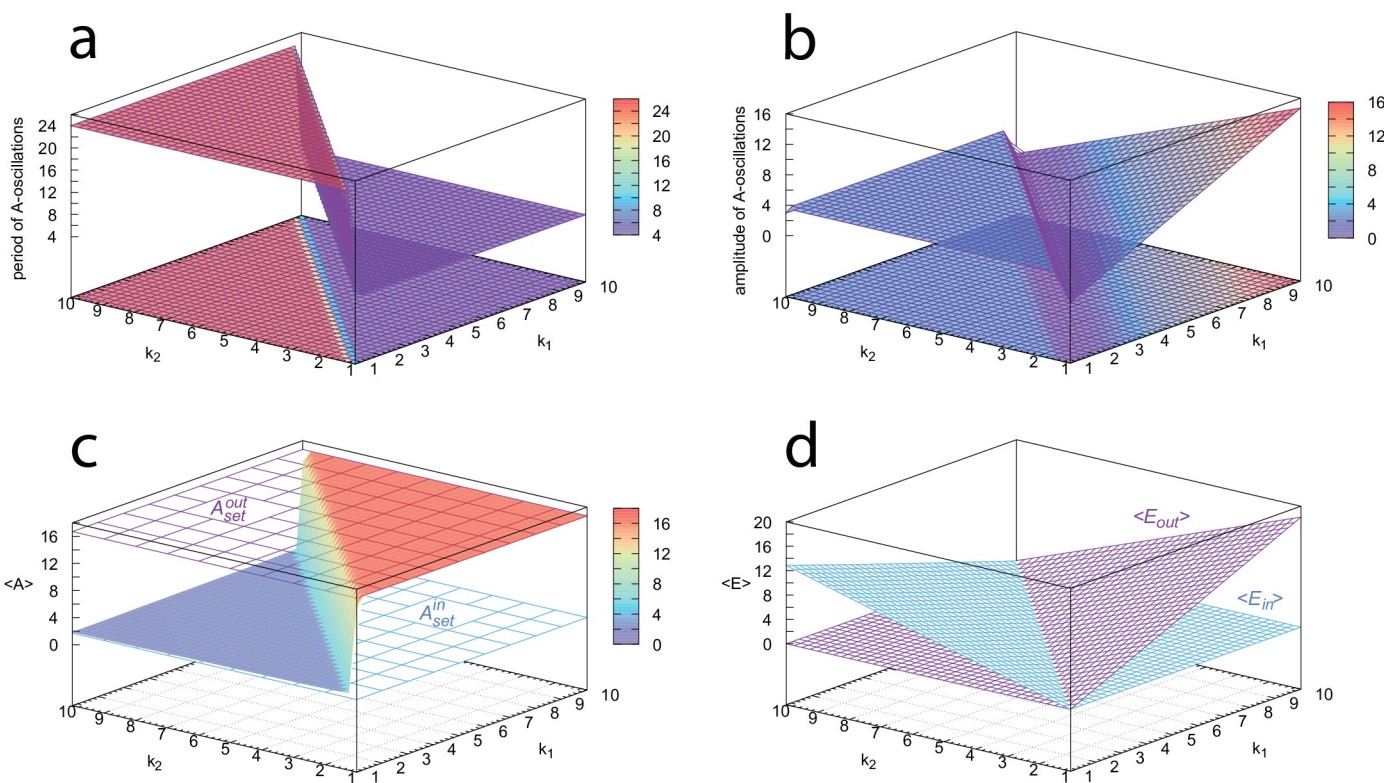

**Fig 6. Overview of the oscillatory properties of combined motifs m3 and m5.** (a) The period switches in dependence whether the inflow or the outflow controller is dominating. (b) Amplitude of the $A$-oscillations. (c) The controllers' $<A>$ values follow closely their set-points. (d) Average values of the oscillatory $E_{in}$ and $E_{out}$ concentrations and their up-regulation in dependence of the perturbations. Rate constants and initial concentrations as for the oscillatory case in Fig 5. All properties were calculated after 500 time units when (oscillatory) steady state conditions were established. See also S2 Gnuplot for an interactive visualization of each panel.

or external environmental disturbances and is involved in cell cycle arrest. p53 is also considered to prevent tumor development by inducing apoptosis in response to DNA-damage and other stress signals [36].

In normal unstressed cells p53 is at low levels due to different proteasomal degradation reactions including ubiquitin-dependent [37] and ubiquitin-independent [38–40] pathways. For the ubiquitin-independent pathways the enzyme NAD(P)H:quinone oxidoreductase 1 (NQO1) has been indicated to have a major regulatory role [41, 42]. In unstressed cells there is further evidence that p53 and the circadian clock [30] undergo cooperative regulations [43–46] via the Per2 protein. Per2 is not only an important component of the human circadian oscillator [47], but also takes part in the input and output pathways of the clock [48]. Under normal (unstressed) conditions p53 has been found to inhibit expression of Per2 by binding to its promotor [43]. Overexpressing Per2 in HCT116 cells resulted in a significant increase in p53 mRNA [43] or in an induced apoptosis in lung cancer cells [49], indicating that Per2 can activate the synthesis of p53. In addition, due to its binding to p53, Per2 has been found to inhibit the Mdm2-mediated degradation of p53 which leads to a stabilization of p53. This Per2-p53 feedback loop has the typical properties of an inflow type of controller and has the same basic structure as the m3 loop in Fig 3. This suggests that p53 is kept by the circadian clock at an acceptable minimum ("preconditioning" [45]) level with set-point $p53_{min}$, which allows a sufficiently rapid up-regulation of p53 in the case of stress/DNA-damage.

In case of stress/DNA-damage p53 is up-regulated by ataxia telangiectasia mutated (ATM) kinase [50, 51]. The treatment of human MCF7/U280 cell lines with 10 Gy gamma radiation showed oscillations in p53 and Mdm2 with a period length of about 5-6 hours [52]. An interesting feature of these oscillations is that their period is relatively stable, while there is a considerable variation in their amplitudes. It has also been pointed out [52] that a significant fraction of the MCF7/U280 cells (about 40% at 10 Gy) do not oscillate, i.e., either showed no variations in p53 or showed only slowly varying fluctuations. In analyzing the p53-Mdm2 negative feedback loop, Jolma et al. [16] found that the loop can show harmonic oscillations when the respective degradations of p53 and Mdm2 approach zero-order. The conservative feature of these oscillations not only could explain the constancy of the period and the stochastic variation in the amplitude, but as a motif 5 outflow controller [18], the set-point of the p53-Mdm2 loop provides an upper p53 concentration limit, probably to avoid a premature apoptosis of cells.

A Fourier analysis of the p53 oscillations [53] showed indeed a major harmonic peak at about 5-6h along with minor 2nd and 3rd-order harmonics at lower periods. The rise of the Fourier transform at higher period lengths (>10h) provides evidence for an additional loop, which Geva-Zatorsky et al. [53] considered to be a feedback loop between ATM and p53. In this feedback loop the active (phosphorylated) form of ATM (ATM*) activates p53 via CHK2 (checkpoint kinase 2) [51, 54], while p53 inhibits ATM* via the activation of phosphatase WIP1 [51, 55, 56]. A closer look at the p53-ATM* loop shows that it acts as a motif m1 ([18]) inflow controller. The inflow control function of this loop suggests that the loop's set-point, $p53_{min}^{stress}$, keeps the p53 concentration in stressed cell at a minimum level, but higher than the set-point imposed by the circadian clock. As we will show below the set-point defined by the ATM* controller increases with the stress level, i.e. shows rheostasis [57], and counteracts perturbations which may accidentally drive p53 to lower levels.

Based on these observations we arrive at a p53 homeostatic model of three interlocked feedback loops with period lengths of p53 oscillations which are dependent on the stress level and the ruling feedback loop responding to it. Fig 7 shows a schematic representation of the model. In unstresssed cells the inflow control properties of the p53-Per2 feedback loop, analogous to motif m3, ensures that p53 is on average at a minimum low level (with set-point $p53_{set}^{Per2}$) compensating for the proteasomal ubiquitin-independent degradations of p53 via NQO1. In stressed cells, several factors increase the level of p53, including its activation by ATM, the inhibition of the ubiquitin-independent degradation pathways of p53, and the stabilization of p53 by chaperones such as HSP90. The inflow control structure between p53 and the activated ATM loop (motif 1) now drives p53 levels up to $p53_{set}^{ATM*}$. As stabilization and concentration of p53 further increases, the Mdm2-p53 control loop (motif 5) will oppose further increase of p53, at least temporarily. However, since the set-point of this controller ($p53_{set}^{Mdm2}$) is given by the ratio between synthesis and degradation rates of Mdm2, the set-point of the Mdm2-controller may further increase when Mdm2 is stabilized by chaperones/HSP90 and the Mdm2 turnover is inhibited [58].

Fig 8 shows the reaction scheme of the model for unstressed and stressed cells. The model consists of 9 coupled rate equations. Dependent on the stress level, reactions outlined in light gray are low in their reaction rates/concentrations, while reactions outlined in black are the dominating ones. For the sake of simplicity we used the single variable *s* to mediate the stress into the network, both for the activation of ATM (with activation constant $K_{as}$) and the inhibition of the NQO1-mediated proteasomal degradation of p53 (with inhibition constant $K_{Is}$). In addition, we also include a stress-related increase of p53 via $k_1$ in parallel to its activation by ATM*. This additional activation may be related due to the presence of reactive oxygen species

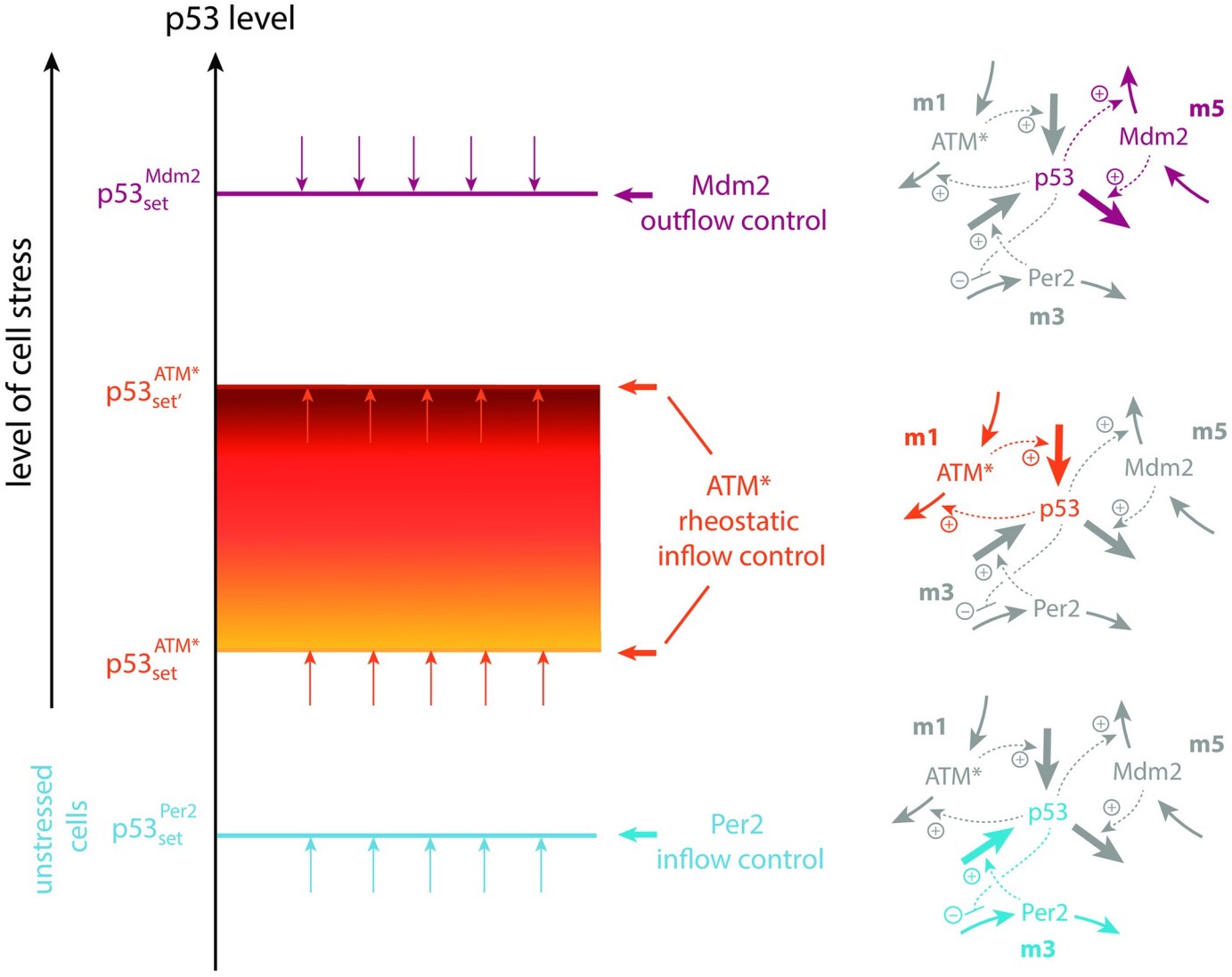

**Fig 7. Hierarchical regulation of p53 by Per2, ATM\*, and Mdm2 in unstressed and stressed cells.** Outlined in blue is the regulation of p53 in unstressed cells, where Per2, acting as an inflow regulator, keeps p53 at a low "preconditioning" level [45]). In the presence of cell stress ATM\* is upregulated (outlined in orange). The rheostatic set-point [57] of this inflow controller increases with increasing cell stress until control by Mdm2 at higher stress levels (outlined in purple) opposes a further increase of the p53 level. Schemes to the right show the color-coded active control loops and the grayed inactive ones.

[59], although the reaction pathways, as indicated by the question mark, are not well understood.

The activation of ATM to ATM\* by the stress level $s$ is described by the rate equation

$$\dot{ATM^*} = \frac{k_{26} \cdot s}{K_{as} + s} - \left( \frac{k_{27} \cdot ATM^*}{k_{28} + ATM^*} \right) \cdot p53 \tag{13}$$

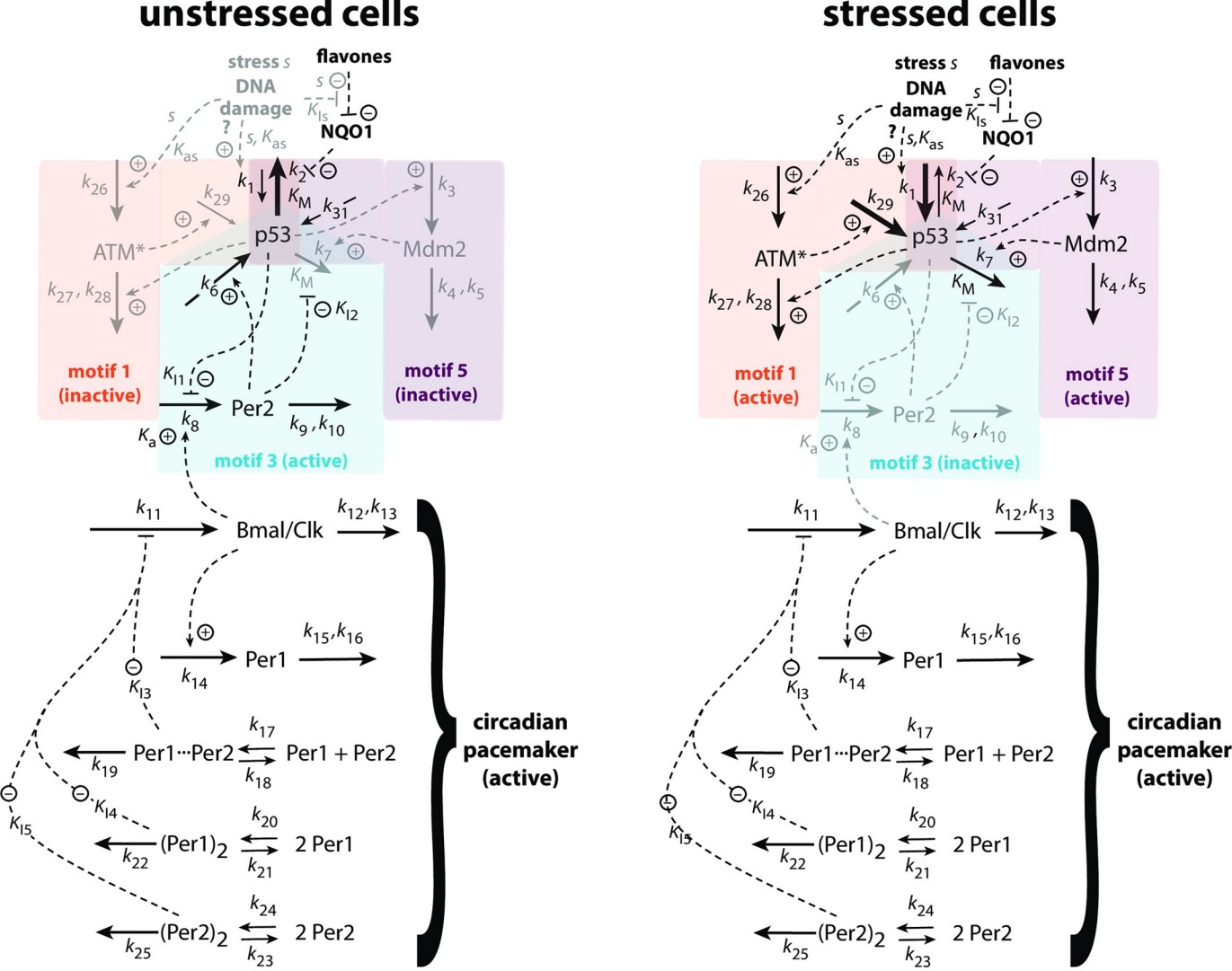

**Fig 8. Model of the three interlocked feedback loops regulating p53.** In unstressed cells (left panel) the m3-loop is active (outlined by the blue area) in which Per2, coupled to the circadian pacemaker, ensures a minimum level of p53. The other controllers (m1 and m5, outlined in gray within their respective orange and purple areas) remain inactive. In stressed cells (right panel) the ATM*-based m1 controller becomes first active (orange area) while the p53-interacting Per2 is downregulated, but without affecting the circadian rhythmicity of Per1 and Bmal/Clk. With increasing cell stress (described by parameter $s$) Mdm2 is upregulated (purple area). The Mdm2 controller opposes an increase of p53 above the Mdm2 set-point, possibly to avoid premature apoptosis. For rate equations, see main text.

The rate equation for $p53$ consists of four inflows and two outflows (Fig 8).

$$\dot{p53} = k_{31} + \frac{k_1 s}{K_{as} + s} - \left(\frac{k_2 \cdot p53}{K_M + p53}\right) \cdot \left(\frac{K_{Is}}{K_{Is} + s}\right) + k_{29} \cdot ATM^* + k_6 \cdot Per2$$

$$- \left(\frac{k_7 \cdot p53}{K_M + p53}\right) \cdot \left(\frac{K_{I2}}{K_{I2} + Per2}\right) \cdot Mdm2 \tag{14}$$

The first term, $k_{31}$, is a constitutive (constant) expression term for p53 [60], while the second and third terms represent, respectively, stress-induced activation of p53 production and a

stress-induced inhibition of the proteasomal ubiquitin-independent degradation of p53 via NQO1.

The remaining rate equations are:

$$\dot{Mdm2} = k_3 \cdot p53 - \frac{k_4 \cdot Mdm2}{k_5 + Mdm2} \tag{15}$$

$$\dot{Per2} = k_8 \cdot \left(\frac{K_{I1}}{K_{I1} + p53}\right) \cdot \left(\frac{Bmal/Clk}{K_a + Bmal/Clk}\right) - \frac{k_9 \cdot Per2}{k_{10} + Per2}$$
$$- k_{17} \cdot Per1 \cdot Per2 + k_{18} \cdot (Per1 \cdots Per2) - 2k_{24} \cdot (Per2)^2 + 2k_{23} \cdot (Per2_2) \tag{16}$$

$$\dot{Bmal/Clk} = k_{11} \cdot \left(\frac{K_{I3}}{K_{I3} + (Per1 \cdots Per2)}\right) \cdot \left(\frac{K_{I4}}{K_{I4} + (Per1_2)}\right) \cdot \left(\frac{K_{I5}}{K_{I5} + (Per2_2)}\right)$$
$$- \frac{k_{12} \cdot Bmal/Clk}{k_{13} + Bmal/Clk} \tag{17}$$

$$\dot{Per1} = k_{14} \cdot Bmal/Clk - \frac{k_{15} \cdot Per1}{k_{16} + Per1} - k_{17} \cdot Per1 \cdot Per2 + k_{18} \cdot (Per1 \cdots Per2)$$
$$- 2k_{20} \cdot (Per1)^2 + 2k_{21} \cdot (Per1_2) \tag{18}$$

$$\frac{d(Per1 \cdots Per2)}{dt} = k_{17} \cdot Per1 \cdot Per2 - (k_{18} + k_{19}) \cdot (Per1 \cdots Per2) \tag{19}$$

$$\frac{d(Per1_2)}{dt} = k_{20} \cdot (Per1)^2 - (k_{21} + k_{22}) \cdot (Per1_2) \tag{20}$$

$$\frac{d(Per2_2)}{dt} = k_{24} \cdot (Per2)^2 - (k_{23} + k_{25}) \cdot (Per2_2) \tag{21}$$

As indicated by Fig 7 p53 is controlled in this model by three feedback loops. Rate parameters have been chosen such that each of the feedback loops has integral control/feedback with defined set-points and, when oscillatory, defined period lengths.

In the absence of stress p53 is rapidly degraded by the proteasome. In this case Per2 acts as an inflow controller with a set-point given by Eq 5, i.e.,

$$p53_{set}^{Per2} = \frac{K_{I1}(k_8 - k_9)}{k_9} \tag{22}$$

Since we assume that the degradation reaction of p53 are zero-order with respect to p53, the Per2 controller oscillates around $p53_{set}^{Per2}$ with a period described by Eq 10, i.e.,

$$P_{p53}^{Per2} = \frac{2\pi}{\sqrt{k_6 k_8 K_{I1}}} (K_{I1} + p53_{min}) \tag{23}$$

The values of $k_6$ (0.7), $k_8$ (1.2), $K_{I1}$ (8.0), and $k_9$ (1.0) have been chosen such that $p53_{set}^{Per2}$ is relatively low, i.e., 1.6. $P_{p53}^{Per2}$ is thereby in the circadian range ($\approx$24h).

Per2, which takes part in the regulation of p53 in unstressed cells, is an important component of the mammalian circadian clock [48]. In Fig 8 we have included a relatively simple

model of the mammalian circadian pacemaker, where Per2 together with Per1 take part in a transcriptional-translational negative feedback loop. In this negative feedback the protein complex between Bmal1 and Clock (Bmal/Clk) activates the transcription of Per1 and Per2. Per1 and Per2, on the other hand, inhibit their own, by Bmal/Clk induced, transcription. By binding to PAS domains [61] homo- and heterodimers between Per2, Per1, and other protein complexes are formed which take part in the inhibition of the transcriptional activity of Bmal/Clk [47]. In the circadian pacemaker part of the model (Fig 8) we included the formation of heterodimers between Per2 and Per1, as wells as the formation of homodimers of Per2 and Per1. In the above equations $(Per1_2)$ and $(Per2_2)$ denote the respective concentrations of the $Per1$ and $Per2$ homodimers, while $(Per1 \cdots Per2)$ denotes the concentration of the $Per1$-$Per2$ heterodimer.

When stress is present, but not too high ($0.1 \leq s \leq 1$), ATM* determines the average concentration of p53 and the frequency of the p53 oscillations. By setting Eq 13 to zero and assuming zero-order degradation of ATM* with respect to ATM* the set-point of p53 determined by this controller is dependent on the stress level $s$, i.e.,

$$p53_{set}^{ATM^*} = \frac{k_{26}}{k_{27}} \cdot \left( \frac{s}{K_{as} + s} \right) \tag{24}$$

Rate parameters $k_{26}$ (90.0), $k_{27}$ (10.0), and $K_{as}$ (3.0) have been chosen such that $p53_{set}^{ATM^*}$ has a maximum value of 9.0 when $s$ is high. This value has been arbitrarily chosen, with the only requirement that $p53_{set}^{ATM^*}$ should be higher than $p53_{set}^{Per2}$. The ATM* controller's period (being a m1 controller) is calculated to be (S2 Text):

$$P_{p53}^{ATM^*} = \frac{2\pi}{\sqrt{k_{27}k_{29}}} \tag{25}$$

Using, rather arbitrarily $k_{29} = 1.0$, the period of the ATM* controller is approximately 2h.

For high stress levels ($s > 1$) the Mdm2 outflow controller keeps p53 at a much higher set-point analogous to Eq 4, i.e.,

$$p53_{set}^{Mdm2} = \frac{k_4}{k_3} \tag{26}$$

Ignoring the influence of noise [16], p53 oscillates now around the set-point described by Eq 26 with period

$$P_{p53}^{Mdm2} = \frac{2\pi}{\sqrt{k_3 k_7}} \tag{27}$$

Using values of $k_3$ and $k_7$ of respectively 3.0 and 0.5 the period of the Mdm2 controller is 5.1h. With $k_4 = 50.0$ the value of $p53_{set}^{Mdm2}$ is 16.6 and defines an upper bound of the p53 concentration. How this upper bound can be further increased and finally may lead to apoptosis will be discussed below.

Fig 9 shows how the steady state period and average levels of p53 change with the stress signal $s$. At certain stress levels the controllers Per2, ATM*, and Mdm2 are individually up-regulated. They defend their set-points and frequencies of the p53 oscillations. Mrosovsky [57] termed the defense of different environmentally-induced set-points as "reactive rheostasis".

In Fig 9a we suggest how a stress-induced inflow to p53 (second term in Eqs 14 and 28) and a stress-induced inhibition of p53-degradation (third term in Eqs 14 and 29) may change with

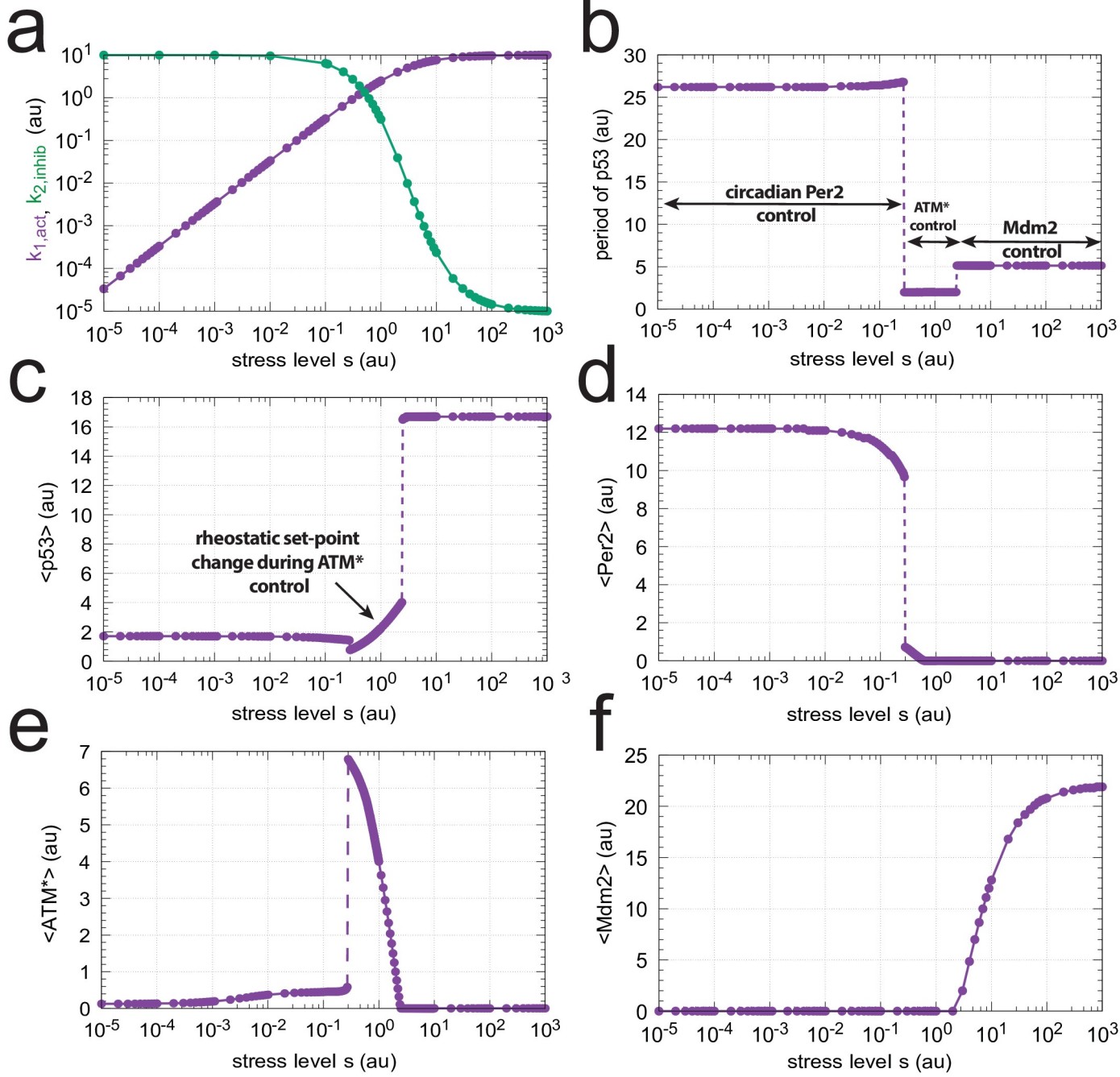

**Fig 9. Change of steady state levels in the model (Fig 8) as a function of stress level *s*.** (a) Change of $k_{1,act}$ (Eq 28) and $k_{2,inhib}$ (Eq 29), (b) p53 period length, (c) average p53 concentration <p53>; inset: same figure, but abscissa (*s*) is linear. (d) average Per2 concentration <Per2>, (e) average ATM* concentration <ATM*>, and (f) average Mdm2 concentration <Mdm2>. Parameter values: $k_1 = k_2 = 10.0$, $k_3 = 3.0$, $k_4 = 50.0$, $k_5 = 1 \times 10^{-6}$, $k_6 = 0.7$, $k_7 = 0.5$, $k_8 = 1.2$, $k_9 = 1.0$, $k_{10} = 1 \times 10^{-6}$, $k_{11} = 0.85$, $k_{12} = 0.7$, $k_{13} = 1 \times 10^{-6}$, $k_{14} = 1.0$, $k_{15} = 0.7$, $k_{16} = 1 \times 10^{-6}$, $k_{17} = k_{18} = 1 \times 10^3$, $k_{19} = 0.0$, $k_{20} = 10.0$, $k_{21} = 0.5$, $k_{22} = 0.0$, $k_{23} = 1 \times 10^6$, $k_{24} = 1 \times 10^3$, $k_{25} = 0.0$, $k_{26} = 90.0$, $k_{27} = 10.0$, $k_{28} = 1 \times 10^{-6}$, $k_{29} = 1.0$, $k_{31} = 1.0$, $K_M = 1 \times 10^{-6}$, $K_{I1} = K_{I3} = K_{I4} = K_{I5} = 8.0$, $K_{I2} = 1 \times 10^6$, $K_{Is} = 3.0$, $K_a = 0.012$, $K_{as} = 3.0$. Initial concentrations: $p53_0 = 1.49$, $Mdm2_0 = 9.84 \times 10^{-8}$, $Per2_0 = 3.82 \times 10^{-2}$, $Bmal/Clk_0 = 1.00$, $Per1_0 = 1.48 \times 10^{-1}$, $Per1\cdots Per2_0 = 5.66 \times 10^{-3}$, $(Per1_2)_0 = 1.82 \times 10^{-1}$, $(Per2_2)_0 = 1.46 \times 10^{-6}$, $ATM_0^* = 8.46$. Steady state levels were recorded after 3000 time units (h).

stress level $s$.

$$k_{1,act} = k_1 \cdot \left( \frac{s}{K_{as} + s} \right) \tag{28}$$

$$k_{2,inhib} = k_2 \cdot \left( \frac{K_{Is}}{K_{Is} + s} \right) \tag{29}$$

Panels b and c show how the individual controllers, dependent on the stress level, determine p53's period length and average level. Panels d-f show the controllers Per2 (d), ATM* (e), and Mdm2 (f) and their abrupt up/downregulation at different stress levels.

Fig 10 shows the steady state oscillations at four different stress levels. Panel a shows the Per2 and p53 oscillations at low/no stress. In agreement with experiments [45] Per2 peaks a couple of hours earlier than p53. As indicated by Fig 9b, 9c and 9d Per2 has control over p53 rhythmicity and its level in unstressed cells. In Fig 10b, at minor stress levels, we see the Per2 and p53 oscillations near the transition to ATM* control, which is indicated by the appearance of short period oscillations in p53 due to the influence of the ATM* controller. Fig 10c shows the oscillations when the ATM* concentration is relatively high (Fig 9e), while in panel d the Mdm2 controller has taken over and is determining the level and period length of the p53 oscillations (see also Fig 9f which shoes the Mdm2 upregulation).

Each of the three controllers, Per2, ATM*, and Mdm2, defend their set-points. As inflow controllers Per2 and ATM* compensate for outflow perturbations, for example by an accidental increase of $k_2$, while the Mdm2 controller will oppose any further increase of p53.

As an example we show the homeostatic/rheostatic behavior of the ATM* controller. The set-point of the ATM* controller, which depends on the stress level $s$ (Eq 24), is defended towards an increase in p53 outflow. Fig 11a shows the behavior of the p53 oscillations when $s = 1.0$ (corresponding to Fig 10c) and $k_2$ undergoes a perturbation at t = 20h from 10.0 to 50.0. The set-point of the controller (2.25) is defended by an upregulation of ATM*, as seen in Fig 11b. Also the period (taken here arbitrarily as 1.99h, Eq 25) is kept constant (Fig 11b). Fig 11c shows the circadian oscillations of Per1 and Bmal/Clk, which are unaffected by the perturbation in $k_2$ and keep a phase relationship in agreement with experimental results [62].

## Synergy conditions for coupled feedback loops

There are certain requirements that need to be met such that a set of coupled negative feedback motifs will cooperate and work together. As pointed out in [18] a cooperative interaction between a set of negative feedback loops will depend on how the set-points of the individual controllers (determined by their $\dot{E}_i$'s) are positioned relative to each other within the concentration space of the controlled variable $A$. Fig 12 shows the rate equations and the corresponding sign structures of the $\dot{E}_i$'s from Fig 1 for the m1, m3, and m5 controllers.

For example, when a m1 and a m5 controller (Fig 13a) are coupled such that $A_{set,1} < A_{set,5}$ the controllers will cooperate and either m1 or m5 will dominate dependent on the perturbation acting on $A$. However, when $A_{set,1} > A_{set,5}$ the two controllers will work against each other, as indicated in Fig 13b. Both controllers are in an "on-state" with the effect that $E_1$ and $E_5$ increase continuously, a situation termed in control engineering as *integral wind-up* [63]. For details, see S3 Text.

The structures of the interacting negative feedback loops shown in Figs 7 and 8 have been taken from the literature (see references cited above). Fig 13c shows the relative setting of the set-points in the p53 concentration space for the Per2, ATM*, and Mdm2 controllers and their

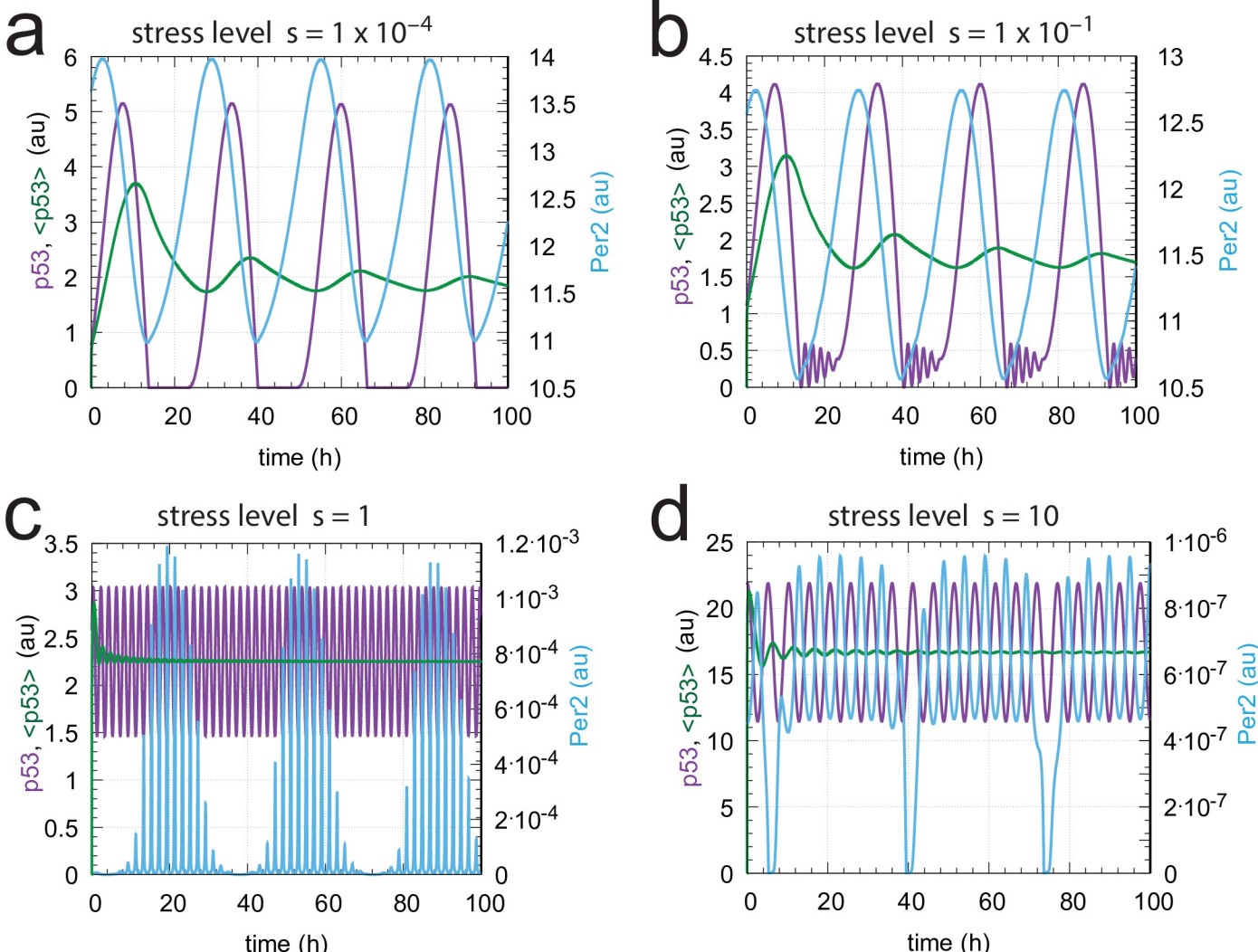

**Fig 10. Steady state oscillations of p53 and Per2, together with <p53>, at different stress levels *s*.** Parameter values are the same as in Fig 9. (a) Oscillatory behavior when $s = 1 \times 10^{-4}$ (low stress level). Per2 determines the state of p53. Initial concentrations: $p53_0 = 7.65 \times 10^{-1}$, $Mdm2_0 = 4.82 \times 10^{-8}$, $Per2_0 = 1.36 \times 10^{1}$, $Bmal/Clk_0 = 0.39$, $Per1_0 = 7.34 \times 10^{-2}$, $(Per1\cdots Per2)_0 = 1.00$, $(Per1_2)_0 = 2.21 \times 10^{-1}$, $(Per2_2)_0 = 1.86 \times 10^{-1}$, $ATM_0^* = 3.92 \times 10^{-10}$. (b) $s = 1 \times 10^{-1}$. The high frequency oscillations of the ATM* controller begin to appear, but the Per2 controller still determines p53 period length. Initial concentrations: $p53_0 = 1.11$, $Mdm2_0 = 7.12 \times 10^{-8}$, $Per2_0 = 1.26 \times 10^{1}$, $Bmal/Clk_0 = 0.47$, $Per1_0 = 8.44 \times 10^{-2}$, $(Per1\cdots Per2)_0 = 1.06$, $(Per1_2)_0 = 2.35 \times 10^{-1}$, $(Per2_2)_0 = 1.58 \times 10^{-1}$, $ATM_0^* = 3.55 \times 10^{-7}$. (c) $s = 1.0$. ATM* is the ruling controller and p53 oscillates with a period of 2h (Eq 25) around the controller's set-point $p53_{set}^{ATM^*} = 2.25$ (Eq 24). Initial concentrations: $p53_0 = 2.58$, $Mdm2_0 = 1.83 \times 10^{-7}$, $Per2_0 = 5.56 \times 10^{-6}$, $Bmal/Clk_0 = 0.17$, $Per1_0 = 3.55 \times 10^{-1}$, $(Per1\cdots Per2)_0 = 1.98 \times 10^{-6}$, $(Per1_2)_0 = 3.03$, $(Per2_2)_0 = 3.10 \times 10^{-14}$, $ATM_0^* = 6.26$. (d) $s = 10.0$. Mdm2 is the dominant controller. p53 oscillates with a period of 5.1h (Eq 27) around a set-point of 16.6 (Eq 26). Initial concentrations: $p53_0 = 21.85$, $Mdm2_0 = 1.17 \times 10^{1}$, $Per2_0 = 4.54 \times 10^{-7}$, $Bmal/Clk_0 = 0.41$, $Per1_0 = 4.03 \times 10^{-1}$, $(Per1\cdots Per2)_0 = 1.83 \times 10^{-7}$, $(Per1_2)_0 = 3.52$, $(Per2_2)_{2,0} = 2.06 \times 10^{-16}$, $ATM_0^* = 4.64 \times 10^{-7}$.

sign structures. It appears encouraging that these feedback loops can be placed in a naturally occurring order in p53 concentration space without wind-up.

## Roles of the individual feedback loops

The spacing of the controllers in Fig 13c suggests that the three feedback loops have certain functions in the regulation of p53. As an inflow controller, the Per2-p53 feedback loop has the

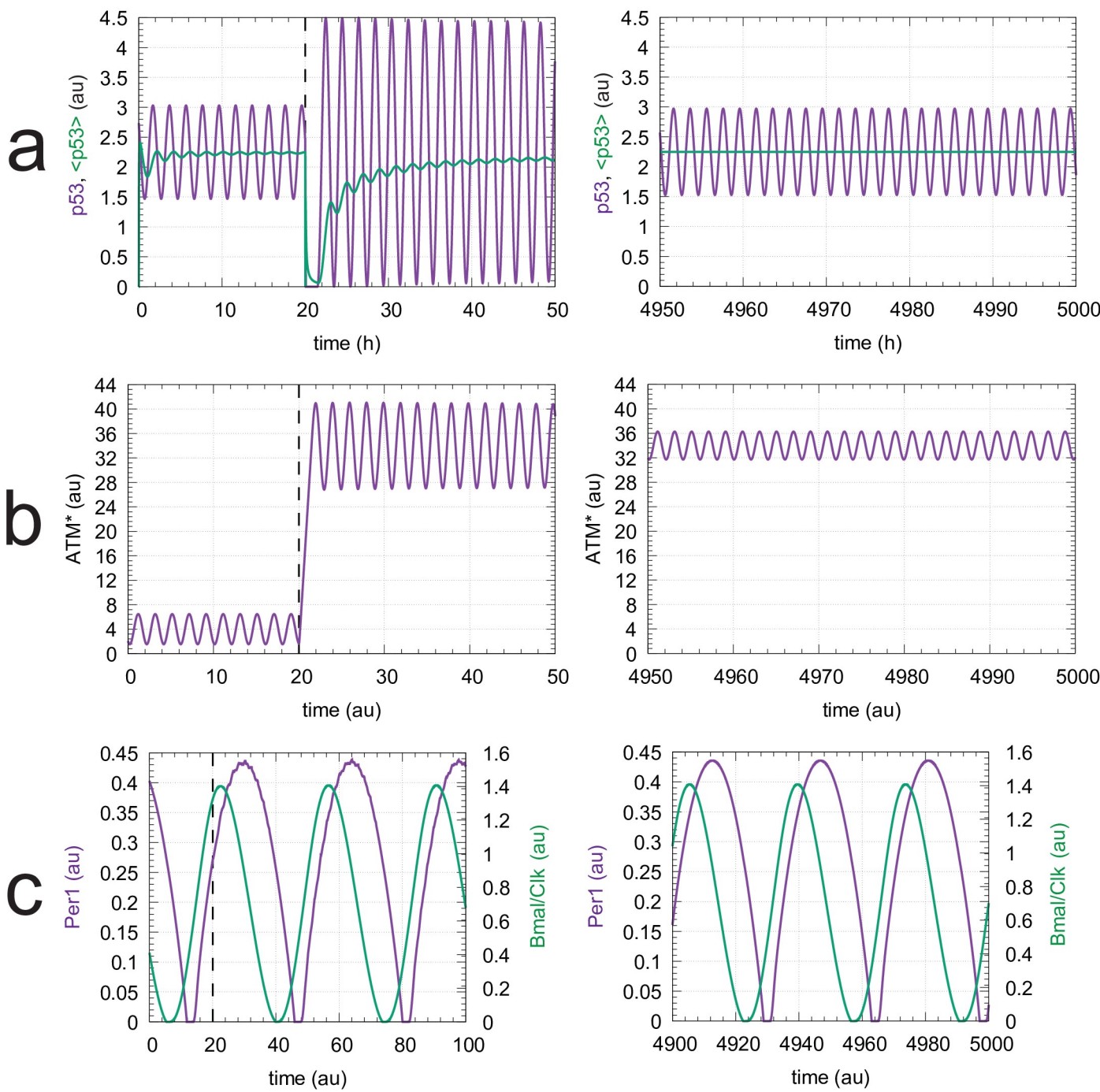

**Fig 11. The ATM* controller defends its *s*-dependent set-points (Eq 24) towards outflow perturbations.** Row a, left panel: p53 oscillations and average p53 levels, <p53>, as a function of time with $s = 1.0$. At time t = 20h $k_2$ is increased from 10.0 to 50.0. Row a, right panel: average p53 concentration is back at the controller's set-point (2.25) with an unchanged period length of 1.99h. Row b, left and right panels: Upregulation of ATM* due to the change in $k_2$. Row c, left and right panels: oscillations in Per1 and Bmal/Clk are unaffected by the $k_2$ perturbation. Initial concentrations and rate parameters as in Figs 10c and 9.

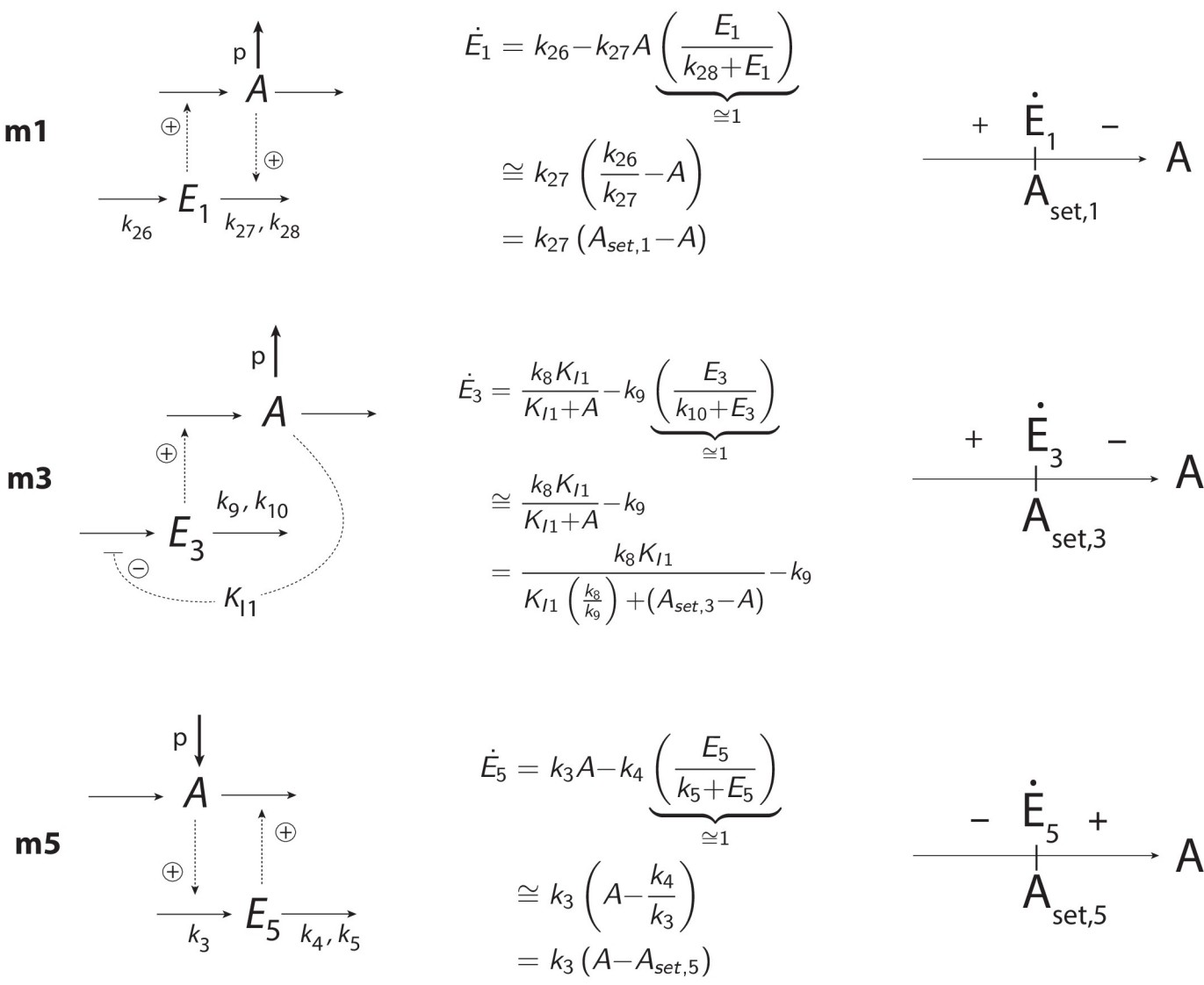

**Fig 12. Rate equations of the $E_i$'s (Fig 1) and their corresponding sign structures $\dot{E}_i$ for the three controllers m1, m3, and m5 with set-points $A_{set,i}$.** Reactions with "p" indicate perturbations. When the controlled variable $A$ is at its set-point ($A = A_{set,i}$) we have that $\dot{E}_i = 0$. When $\dot{E}_i > 0$ the controllers are active, i.e. inflow controllers m1 and m3 add more $A$ to the system, while outflow controller m5 removes $A$ from the system. When $\dot{E}_i < 0$ controllers become inactive and the $E_i$'s and their compensatory fluxes go to zero/low values.

apparent function to keep p53 in unstressed cells at a certain minimum level in order to allow a sufficiently rapid p53 upregulation [44] in case DNA damage occurs.

In case of DNA damage, the ATM*-p53 loop, is up-regulated. This loop keeps p53 at a higher set-point dependent on the stress level $s$. The ATM* controller defends its set-point towards increased degradations of p53 as long as stress is encountered. This suggests that as long as DNA-stress is present, the ATM*-loop ensures that p53 is not decreased due to stress-unrelated or accidental degradations of p53. In a way the ATM*-loop acts as a one-way concentration valve, not allowing that p53 concentrations are decreased below a certain minimum level. The stress-dependent increase of the p53 set-point by the ATM*-p53 loop is a nice example of what Mrosovsky [57] has termed *rheostasis*. Rheostasis is defined as a homeostatic

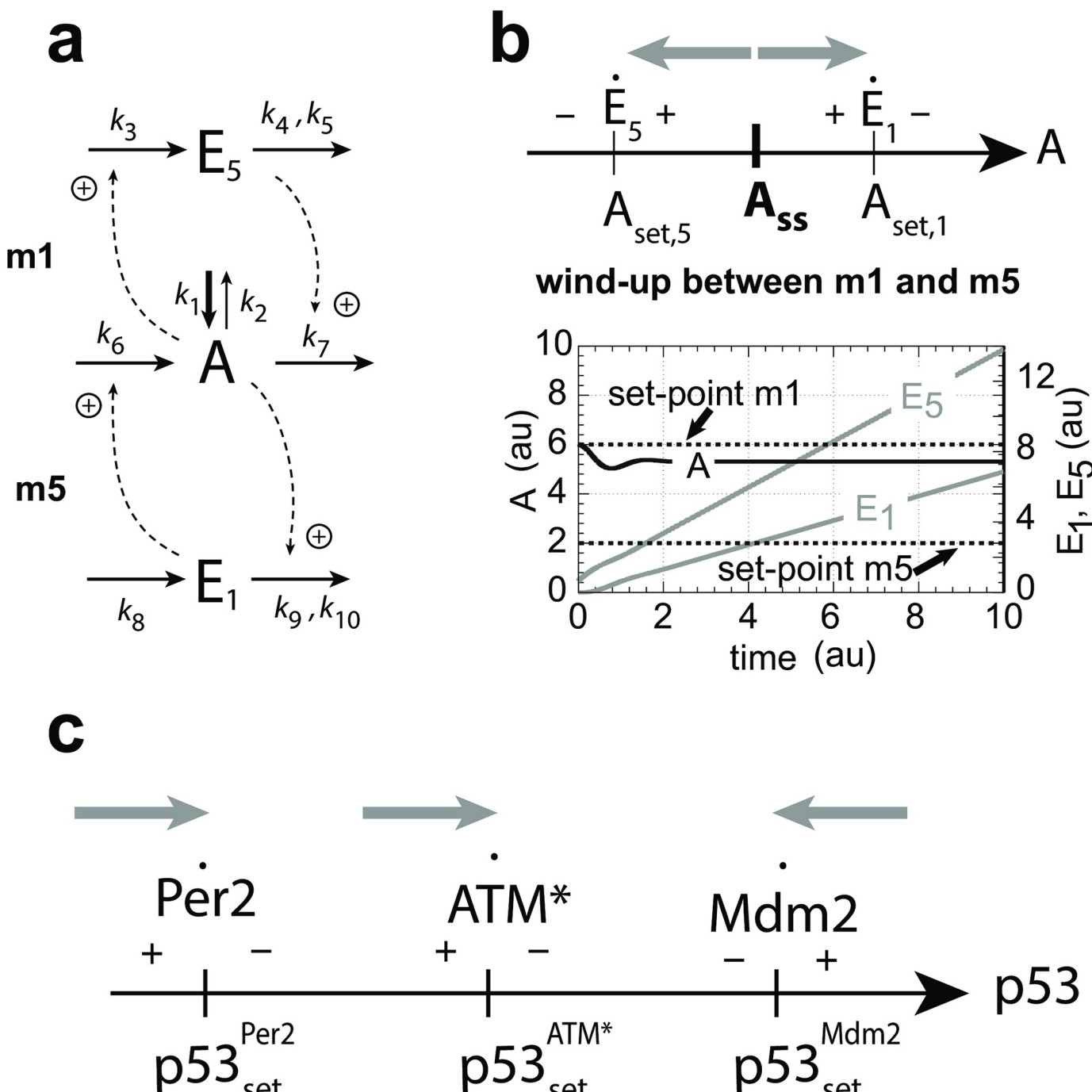

**Fig 13. Cooperative and dysfunctional wind-up behavior in combined negative feedback loops.** (a) Combined controllers m1 and m5 (S3 Text). (b) Wind-up behavior when $A_{set,1} > A_{set,5}$. Because $\dot{E}_1$ and $\dot{E}_5$ meet each other in this case with positive signs, each controller pulls in the direction of its own set-point (gray arrows). As a result, $E_1$ and $E_5$ increase continuously and the controlled variable $A$ lies somewhere between the two set-points dependent on the individual aggressiveness [18] of m1 and m5. S3 Text shows in addition the cooperative behaviors of the two controllers when set-points are switched. (c) Cooperative behavior of the Per2, ATM*, and Mdm2 controllers with respect to p53 regulation. Gray arrows show the direction in p53-concentration space into which each controller pulls.

system when the set-point is changed but defended in relationship to a changed external or internal environmental condition or due to stress. A typical example of rheostatic regulation is the defended increase of body temperature (fever) due to an infection. For more examples of rheostasis, see Ref [57].

As indicated earlier [16], the role of the Mdm2-p53 loop seems to avoid premature apoptosis by counteracting uncontrolled rises of p53 above the Mdm2-determined set-point. However, the set-point of the Mdm2-p53 controller does not seem to be fixed. Peng et al. [58] found that treating DLD1 cells with the DNA-damaging agent camptothecin (CPT) led to a decrease in Mdm2 levels. The authors' interpretation was that Mdm2 degradation under DNA-stress is actually promoted. Such an increase in the Mdm2 degradation ($k_4$) by DNA-damaging conditions would lead to an increased p53 set-point of the Mdm2 controller (Eq 26), which ultimately could reach apoptotic p53 levels. Chaperones and HSP90 lead to an additional stabilization of p53.

Thus, our model suggests that the individual feedback loops act as temporary stabilizers of p53 when DNA stress is encountered. They result in a gradual step-wise increase in p53 concentration, where each step is under homeostatic (rheostatic) control. When DNA repair is successful and stress levels are removed, p53 concentration falls back to its minimum set-point determined by the Per2-p53 loop. In principle, additional, not yet identified negative feedback loops of p53 with other controllers could be involved in such a rheostatic regulation of p53 during DNA stress. Considering the "plethora of proposed feedback interactions" of p53 [64], an investigation of additional feedbacks in terms of their inflow/outflow behavior may provide further insights and novel suggestions about the workings between different controllers in the p53 network.

## Why oscillations?

The here presented homeostatic (rheostatic) model on how p53 levels are controlled does not necessarily need to involve oscillations. As shown in Fig 5 both the oscillatory and the non-oscillatory versions of the coupled homeostats work equally well. The same goes for the p53 system (Fig 9) when the ATM* and Mdm2 controllers are in a non-oscillatory mode (S4 Text). Thus, sustained or damped oscillations could simply be a byproduct of the negative feedback loops. What supports partly such a view is that a large fraction of $\gamma$-irradiated cells ($\approx 40\%$) do not show oscillations [52] and that there is a considerable heterogeneity of p53 dynamics even in genetically identical cells [64]. The various proteasomal p53 degradation pathways, which have been discovered [37–39], may provide an explanation why some cells show oscillations while others don't. The pathways may differ in their binding strength between p53 and the different proteasomal proteins. In non-oscillatory cells the proteasomal degradation pathway may have a weak binding between p53 and pathway proteins/enzymes (which determine the rate of p53 degradation), and which would lead to a high overall $K_M$ and to approximately first-order p53 degradation kinetics. In oscillatory cells, on the other hand, the dominant proteasomal degradation pathway may have a tighter binding between p53 and proteasomal proteins with the result of overall lower $K_M$ values and close to zero-order degradation kinetics. As long as ATM* and Mdm2 contain integral feedback loops one would expect the same regulatory outcome independent of the p53-degradation kinetics, i.e. whether feedback loops are oscillatory or not. The large heterogeneity of p53 dynamics seems to indicate that there is no selective advantage whether homeostatic control of p53 in some instances occurs oscillatory while under other conditions it does not. Porter et al. found that physiologically relevant DNA damage responses apparently begin already after very few p53 pulses or even before the first p53 pulse is completed, and that coordination of p53 target genes increases with successive

p53 pulses [65]. This observation can be interpreted in such a way that the coordination of p53 target genes become established when p53 approaches a stress-level dependent steady state/ set-point with the suggested [51] possibility that p53 pulses and their dynamics trigger different signal transduction pathways.

There is also the possibility that some controllers (for example Per2 and Mdm2) are oscillatory (due to zero-order degradation with respect to p53), while the ATM$^*$ feedback loop is non-oscillatory (due to first-order degradation of p53). This would explain the observation that certain cells, by being less susceptible towards gamma irradiation, are controlled by ATM$^*$ and thereby are non-oscillatory, while in other cells, which are more sensitive towards irradiation, p53 is controlled by Mdm2 and shows oscillations.

## Summary and conclusion

We have shown that oscillatory homeostats can impose specific frequencies on the oscillations of a controlled variable. In case of perturbations or stress acting on the controlled variable a switch between one oscillatory controller to another is accompanied by a corresponding switch in frequency through the controlling feedback. By analyzing the inflow/outflow control structures of three p53 negative feedback loops (Per2, ATM$^*$, and Mdm2) we were able to assign certain functionalities to each of them. Per2 provides circadian inflow control over p53 by keeping it at the lowest set-point level, ensuring that p53 can be rapidly up-regulated in case of DNA damage/stress. In case of DNA damage the ATM$^*$-p53 feedback loop, another inflow controller, leads to increased p53 levels depending on the stress level. Since the ATM$^*$-induced p53 concentrations are under homeostatic control and defended, the ATM$^*$-p53 feedback loop provides a nice example of what Mrosovsky has termed rheostasis. As an outflow controller, the Mdm2-p53 loop does not allow that p53 levels are raised above the controller's set-point, probably to avoid premature apoptosis. However, additional mechanisms, such as chaperones and heat shock proteins, in particular HSP90, seem to increase Mdm2's set-point and stabilize p53 at a higher level, which finally may lead to apoptosis.

We have considered here only negative feedback loops without the addition of feedforward or positive feedback. While the circadian rhythms show limit-cycle behavior, the p53/ATM$^*$ and p53/Mdm2 oscillations are conservative. Including positive feedbacks into the model will certainly make changes in the network's dynamics and may lead to limit cycle behavior where conservative oscillations are presently observed. However, we do not think that the qualitative (homeostatic) properties of the interacting negative feedback loops will be significantly altered. Although the p53 model presented here is a far cry from what seems to go on in a real cell, the inflow/outflow approach and the conditions how negative feedback loops can interact without dysfunctional behavior (wind-up) appears to be an alternative and novel aspect how to analyze the organization of feedback loops in cells and organisms.

## Supporting information

**S1 Matlab. Matlab programs.** A zip-file with Matlab programs showing the results from Figs 5a, 5b, 10a–10d and 11a–11c (left panels).
(ZIP)

**S1 Gnuplot. Interactive visualization.** A zip-file with a gnuplot script and an avi video file showing an interactive view of Fig 4.
(ZIP)

**S2 Gnuplot. Interactive visualization.** A zip-file with a gnuplot script and avi video files showing interactive views of panels a-d in Fig 6.
(ZIP)

**S1 Text. Harmonic approximation of period lengths and amplitudes of oscillatory m3 and m5 controllers.**
(PDF)

**S2 Text. Determination of set-point and period length of the ATM* controller.**
(PDF)

**S3 Text. Combined m1 and m5 controllers and their cooperative and wind-up behaviors.**
(PDF)

**S4 Text. Steady state levels of the model as a function of stress level *s* when feedback loops are non-oscillatory.**
(PDF)

## Author Contributions

**Conceptualization:** Peter Ruoff, Nobuaki Nishiyama.

**Investigation:** Nobuaki Nishiyama.

**Methodology:** Peter Ruoff.

**Validation:** Peter Ruoff, Nobuaki Nishiyama.

**Visualization:** Peter Ruoff.

**Writing – original draft:** Peter Ruoff.

**Writing – review & editing:** Peter Ruoff.

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
