## [Decision Letter · Decision Letter 0]

5 Feb 2020

PONE-D-19-35675

Frequency switching between oscillatory homeostats and the regulation of p53

PLOS ONE

Dear Ruoff,

Thank you for submitting your manuscript to PLOS ONE. After careful consideration, we feel that it has merit but does not fully meet PLOS ONE’s publication criteria as it currently stands. Therefore, we invite you to submit a revised version of the manuscript that addresses the points raised during the review process.

Both reviewers are in agreement that the manuscript is in need of major revisions.  The reviewers provide multiple suggestions for modifications to figures, text (rationale, assumptions, etc), and mathematical equations that will greatly improve clarity.  In particular, the reviewers note several places where the writing could be improved to eliminate ambiguities and inaccuracies.  Another example of this issue comes from the abstract: “Mdm2 on its side keeps p53 at a maximum level to avoid premature apoptosis”.  I encourage the authors to not only respond to the reviewer feedback but also carefully review the entire text before submitting a revised manuscript.

We would appreciate receiving your revised manuscript by Mar 21 2020 11:59PM. To enhance the reproducibility of your results, we recommend that if applicable you deposit your laboratory protocols in protocols.io, where a protocol can be assigned its own identifier (DOI) such that it can be cited independently in the future. For instructions see: http://journals.plos.org/plosone/s/submission-guidelines#loc-laboratory-protocols

We look forward to receiving your revised manuscript.

Kind regards,

Robert S. Weiss

Academic Editor

PLOS ONE

Reviewers' comments:

Reviewer's Responses to Questions

**Comments to the Author**

1. Is the manuscript technically sound, and do the data support the conclusions?

Reviewer #1: Partly

Reviewer #2: Yes

2. Has the statistical analysis been performed appropriately and rigorously? 

Reviewer #1: Yes

Reviewer #2: N/A

3. Have the authors made all data underlying the findings in their manuscript fully available?

Reviewer #1: Yes

Reviewer #2: Yes

4. Is the manuscript presented in an intelligible fashion and written in standard English?

Reviewer #1: Yes

Reviewer #2: No

5. Review Comments to the Author

Reviewer #1: Summary:

Peter Ruoff and Nobuaki Nishiyami present a novel model of p53 oscillations in terms of three interlocked negative feedback loops involving Per2 (Circadian Oscillations), ATM* (Radiation Damage Sensing), and Mdm2 (p53 ubiquitinoylation and degradation). The work further develops their theoretical paradigm cited in [16], [17], [18], [21], [25], [27], [28], and [53]. The work also builds upon their biological model previously presented in [16] and [28]. The key findings of this work include demonstration of a multi-step response controlled by the three competing rheostats set by Per2, ATM*, and Mdm2 that is also associated with frequency switching from p53 oscillations associated with Circadian rhythms to faster cycling. The authors frame why oscillations occur in terms of integral control and develop criteria to determine whether interlocked controllers will perform cooperatively, leading to optimal control, or competitively, leading to integral windup.

The papers raised some interesting points but the analyses need to address some important points:

Major Concerns:

Define Type 3, Type 1, and Type 5 controllers in the introduction. This terminology could be confusing because, as an example, Uri Alon’s Intro to Systems Biology describes totally different motifs with the same 1-8 notation. It’s possible that other sources could also define these differently. Further, these motifs only apply to two species systems.

Figure 2 is not intuitive and notationally confusing. In the current configuration, it looks like there is interconversion between A and E_in since k_1,k_2 look like a reversible reaction between them. Change this to something like the empty set ∅ to indicate generation and degradation. Further, you use the Michaelis/Saturation constant K_M multiple times. Denote these as K_(M,i) where i indicates the reaction number. E_out degradation, you list rates k_4 and k_5, but from equation 2, k_5 appears to be a Michaelis constant, similarly with k_10. What does K_I1 denote? Why are letters being used instead of numbers at this point? State that K_M will be omitted in the first model but will be included in later, more complex models.

The authors rely significantly on self-citations notably 16, 18, and 28. Please acknowledge that these are self-citations (e.g. “In [16], we previously derived…”). This would also clarify why the authors refer to the motifs using such specific terminology which might be unfamiliar to the reader as commented on in major revision 1.

Graphically, many of the figures appear to be low resolution. Plots are presented as 3D surfaces that are impossible to read rather than pseudo-colormaps. Other figures appear inappropriately vectorized and show distortions upon scaling. Further, the notation and depiction of reaction networks in Figures 2 and 7 is extremely confusing.

Based on Reference 16, the period derived there is P=2π/(k×k_adapt ) (just below Eq. 5 in Ref 16). Unless I am missing something in that source, why does the equation for period involve a square root as if a geometric average was taken? Is there an error in either this manuscript or the author’s previous article?

Further, in their derivations the authors assume that K_M,K_M^'≪A, this negates the need to even use a Michaelis-Menten form since the reactions would become pseudo-zero order.

Figure 5 showed fluctuations of A. You were previously able to provide reasonable estimates of the average values of 〈A〉,〈E_out 〉,and 〈E_in〉. Can you analytically estimate or predict the magnitude of fluctuations in this figure?

Amend Figure 6 with an idealized/simplified schematic of Figure 7 showing how all 3 feedback motifs affect P53 similar to Figure 2.

Amend Figure 7 with boxes showing which parts constitute the Motif 1, 3, and 5 feedback systems.

No justification of the rate parameters is provided besides that they exist in a region of parameter space expected to provide oscillatory, integral control. In the methods section, please clearly cite previous models and parameter sets from which this current model was constructed.

As opposed to intrinsic stochasticity in the gene circuit, I think there is the question of parameter heterogeneity. Given that 40% of cells treated with the same radiation dose do not respond in an oscillatory fashion, I feel like a Monte Carlo parameter sweep (even if deterministic integration ignoring intrinsic noise is employed) of certain critical parameters like k_30 should be performed in order to investigate the bifurcation behavior of oscillatory/non-oscillatory cells and to provide potential mechanisms for why this has been experimentally observed.

Figure 11 needs to be significantly reworked for clarity. This is the first time all 8 configurations are introduced in the paper in spite of the fact that the authors expect the reader to be familiar with all of them in the introduction. Further, the sign structure arguments are confusing. It may actually be easier for the author to specify the differential equations and the set point for each configuration so that the reader can algebraically verify the sign structure.

In Figure 12c, I understand why a signal less than p53-max stress would decrease to p53-min stress since both signs are negative. However, it seems like Per2 would pull the p53 level towards the homeostatic condition while ATM* would pull it to the min damage condition. Why isn’t this an example of integral windup?

Minor Revisions:

Approximate ≈ or limit → should be used in Equations 4 and 5 since they are subject to the approximations/limiting arguments described in the preceding paragraph. In Reference 16, which was your own previous paper, you make this distinction.

Figure 3 may be more legible as a 2D pseudo color plot with all 3 side by side. 3D surfaces tend to be harder to read. Also, superimposing two 3D surfaces cannot be read.

Figure 5 should be revised as suggested for Figure 3 in minor revision 2. You already appear to have the pseudo-color plots beneath the surfaces. Therefore, you should just use the 2D pseudo-colors.

Grammatical error line 138, change to “degradation of p53, thereby stabilizing p53.”

Don’t denote the radiation damage by k_30. This makes it easily confusable as a reaction rate.

Don’t denote Michaelis constants with the same notation as rate constants.

Grammatical error line 177. Change to “In stressed cells, several…”

The graphical resolution of Figure 8 looks low quality. It seems like things were scaled up inappropriately.

Throughout Figure 8, the resolution of the graphs near the phase transition is insufficient. While the Circadian and high stress states are oversampled, the region around the two phase transitions could benefit from enhanced simulation resolution.

Reviewer #2: The author proposes a dynamical model of the p53 regulation, accounting for three interlocked feedback loops involving p53, Per2, ATM and Mdm2, to illustrate frequency switches that can occur in such type of homeostatic systems in response to environmental signals. They use a structural approach in terms of the analysis of specific motif interplays to interpret the behavior of such systems combining different so called “inflow” and “outflow” controllers.

Although the use of the controller’s theory applied to biological systems to decipher their dynamical behavior is interesting, I believe that the manuscript needs major revisions before being accepted in PloS One. The following major concerns should be addressed before considering the manuscript for publication:

1- The authors should rework in depth the structure and the writing of the manuscript. The biological data used to build the model, the assumptions made, and the analysis of the model outcome should be more clearly delineated. Moreover, some parts of the paper resembles a catalogue of descriptions (for example p.9) which make the manuscript unclear and hard to follow. The description of the model and its analysis are not separated clearly enough, which makes the outcome of the work confusing.

2- Some statements need more explanations/justifications. For example, there is a lack of explanations of

the type of functions (Michaelis-Menten, linear, product of Michaelis-Menten,…) used to model the processes appearing in the differential equations. Moreover, some biological data used to built the model is lacking (role of Bmal/Clk, dimerization of Per1 and Per2).

3- The level of English should be strongly improved. For example: “(K_m not used)” l.81; “we arrive at” l.171 ; “the other rate equations are” l.205; “has apparently the function” l.291,...

6. PLOS authors have the option to publish the peer review history of their article (what does this mean?). If published, this will include your full peer review and any attached files.

Reviewer #1: No

Reviewer #2: No

---

## [Author Response · Author response to Decision Letter 0]

26 Mar 2020

please see attached file "reply_reviewers.pdf"

---

## [Decision Letter · Decision Letter 1]

5 May 2020

Frequency switching between oscillatory homeostats and the regulation of p53

PONE-D-19-35675R1

Dear Dr. Ruoff,

We are pleased to inform you that your manuscript has been judged scientifically suitable for publication and will be formally accepted for publication once it complies with all outstanding technical requirements.

With kind regards,

Robert S. Weiss

Academic Editor

PLOS ONE

Additional Editor Comments (optional):

Both reviewers found the manuscript to be significantly improved, although the two reviewers differed somewhat on their overall assessment.  Reviewer 2 raised a few issues with the revised manuscript that follow on comments from the original submission.  My assessment is that the revised publication does meet the specific PLOS ONE criteria for publication, with the suggestions of reviewer 2 being reasonable but primarily stylistic in nature.

Reviewers' comments:

Reviewer's Responses to Questions

**Comments to the Author**

1. If the authors have adequately addressed your comments raised in a previous round of review and you feel that this manuscript is now acceptable for publication, you may indicate that here to bypass the “Comments to the Author” section, enter your conflict of interest statement in the “Confidential to Editor” section, and submit your "Accept" recommendation.

Reviewer #1: All comments have been addressed

Reviewer #2: (No Response)

2. Is the manuscript technically sound, and do the data support the conclusions?

Reviewer #1: Yes

Reviewer #2: Yes

3. Has the statistical analysis been performed appropriately and rigorously? 

Reviewer #1: Yes

Reviewer #2: N/A

4. Have the authors made all data underlying the findings in their manuscript fully available?

Reviewer #1: Yes

Reviewer #2: Yes

5. Is the manuscript presented in an intelligible fashion and written in standard English?

Reviewer #1: Yes

Reviewer #2: No

6. Review Comments to the Author

Reviewer #1: The authors have addressed all of my questions thoroughly. The figures they altered are much clearer in terms of breaking down the controller motifs.

Reviewer #2: The author proposes a dynamical model of the p53 regulation, accounting for three interlocked feedback loops involving p53, Per2, ATM and Mdm2, to illustrate frequency switches that can occur in such type of homeostatic systems in response to environmental signals. They use a structural approach in terms of the analysis of specific motif interplays to interpret the behavior of such systems combining different so called “inflow” and “outflow” controllers.

Compared to the previous version of the manuscript, there have been substantial efforts to complete and improve the manuscript and a slight effort to improve its clarity with the incorporation of a subsection in the Material and Methods highlighting the outline of the paper. Moreover, some lacking biological data used to build the model (role of Bmal/Clk, dimerization of Per1 and Per2) are now described in the manuscript.

However, I still think that the manuscript needs some major revisions before being accepted in PloS One. The following major concerns should be addressed before considering the manuscript for publication:

1. I’m a bit confused concerning the sections entitled “Cannon's definition of homeostasis and its realization by inflow and outflow controllers. The non-oscillatory case” and “Oscillatory control mode” as it does not clearly appear to me whether it corresponds to an original outcome of the work or a recall of previous works on the realization of homeostasis by inflow and outflow controllers. The authors should clarify this point. If it is just a recall of previous works, I would strongly suggest to move this part to supplementary materials (which would moreover clarify the outcome of the work), with references to this part in the rest of the manuscript whenever needed.

2. The authors should still improve the structure and the writing of the manuscript. The description of the model, the hypothesis and choices made (notably for the parameter values), the results and its analysis are still not separated clearly enough, making the outcome of the work still confusing. Again, some parts of the paper are more like a “catalogue” of descriptions and results, making the manuscript hard to follow. Maybe additional subsections should be considered to clarify the manuscript, for example: a part devoted to the description of the p53 model (biological background, mathematical model proposed, biological data on which the model rely); another on the hypothesis and choices made in particular for the setting of the parameters; and another one focusing on the main results with the most relevant simulation results. I would also suggest to move details of mathematical calculations in supplementary material.

3. There is still a lack of explanations/justifications of the type of functions (Michaelis-Menten, linear, product of Michaelis-Menten,…) used to model the processes appearing in the differential equations.

7. PLOS authors have the option to publish the peer review history of their article (what does this mean?). If published, this will include your full peer review and any attached files.

Reviewer #1: Yes: Xiling Shen

Reviewer #2: No

---

## [Editor Report · Acceptance letter]

6 May 2020

PONE-D-19-35675R1 

Frequency switching between oscillatory homeostats and the regulation of p53 

Dear Dr. Ruoff:

I am pleased to inform you that your manuscript has been deemed suitable for publication in PLOS ONE. Congratulations! Your manuscript is now with our production department. 

With kind regards,

on behalf of

Dr. Robert S. Weiss 

Academic Editor

PLOS ONE